## Registered report

psychology

retrieval practice, testing effect, transfer effect, episodic memory, education

**Authors for correspondence:**
Jade S. Pickering
e-mail: jadespickering@gmail.com
Aidan J. Horner
e-mail: aidan.horner@york.ac.uk

# Retrieval practice transfer effects for multielement event triplets

Jade S. Pickering[1], Lisa M. Henderson[1,2] and Aidan J. Horner[1,2]

[1]Department of Psychology, University of York, York YO10 5DD, UK
[2]York Biomedical Research Institute, University of York, York, UK

 JSP, 0000-0002-7242-9207; LMH, 0000-0003-3635-2481; AJH, 0000-0003-0882-9756

Retrieval practice (RP) leads to improved retention relative to re-exposure and is considered a robust phenomenon when the final test conditions are identical to RP conditions. However, the extent to which RP 'transfers' to related material is less clear. Here, we tested for RP transfer effects under conditions known to induce integration of associated material at encoding, which may make transfer more likely. Participants learned multielement triplets (locations, animals and objects) and one pairwise association from each triplet was tested through RP, re-exposed, or not re-exposed (control). Two days later participants completed a final test of all pairwise associations. We found no evidence for an RP effect compared to re-exposure, but both tested/re-exposed pairs were better remembered than the not re-exposed control condition. We also found that transfer occurred from both tested to untested and re-exposed to not re-exposed pairs. Our results highlight that RP *and* re-exposure can boost retention for directly tested/re-exposed event pairs and associated but untested/not re-exposed event pairs, suggesting re-exposure of integrated information can be of pedagogical value. The results also question the boundary conditions for an increase in retention for RP relative to re-exposure, highlighting the need for a better theoretical understanding of RP effects.

## 1. Introduction

Promoting long-term retention of newly learnt material is a critical aim in education. Experimental psychology has revealed several effective learning strategies that promote retention [1,2]. One such strategy is the retrieval practice (RP) effect [3]. Also known as the test-enhanced learning or the testing effect, the RP effect refers to increased retention of learned information following a retrieval test on the material. RP is claimed to actively

contribute to learning over and above simple re-exposure to, or re-study of, the learned material, and has additional long-term benefits for retention [4,5]. The underlying mechanisms of RP effects are less clear, but one proposal suggests that practicing active retrieval creates a more elaborate memory trace and additional retrieval routes which in turn increases the likelihood of future retrieval [5] (but for alternative accounts, see also e.g. [6,7]). Critically, RP is a well replicated phenomenon, producing meaningful long-term learning effects relative to re-exposure in both the laboratory and the classroom particularly when repeated at spaced intervals [8–12]. A delay between RP and final test of one day to one week optimizes the RP effect (whereas shorter delays can result in final test performance that is equivalent to re-study [11,12]) and so end-of-lesson RP quizzes can be particularly beneficial in the classroom to consolidate just-learned information and facilitate retrieval in future classes or exams [6,10–12]. RP is, therefore, recommended in several educational resources [1,2].

## 1.1. Retrieval practice transfer effects

Despite RP being a highly recommended practice in education, there are still many research questions to resolve before it can be optimized for the classroom. For example, most research into the effect has tested the same information during RP and at final test. Ideally, RP would benefit the learning and retention of not only the information specifically tested, but also related information (e.g. material that is semantically related, or learnt in the same spatio-temporal context). If RP is only useful for the material that is tested, and in the same format that it is tested in, this may limit its generalizability and in turn its pedagogical utility. Recent research has, therefore, examined whether RP produces so-called 'transfer effects'—where performance increases are seen despite changing aspects of retrieval (e.g. the task and material) between RP and final test [13]. Such research has provided evidence for both 'near' transfer such as across test formats between RP and final test (e.g. from cued recall during RP to multiple choice at test) [14,15], and 'far' transfer such as inference questions and problem-solving skills (e.g. a medical student applying previously learned information to form a medical diagnosis) [13]. Thus, RP is a rare case in experimental psychology where 'far' transfer can occur. However, there are situations where transfer appears less robust (see [13] for a review and meta-analysis). For example, although RP transfer has been seen between strongly semantically related prose content [16–19], partially related or unrelated content learnt in the same spatio-temporal context does not appear to show transfer between RP and a final test one to two weeks later [15,20].

More recently, laboratory-based experiments using more tightly controlled stimulus sets have been used to precisely manipulate stimulus–response overlap between RP and final test. Stimulus–response transfer effects refer to when stimulus and response (A–B) are presented initially together, followed by RP for A–? which subsequently increases the probability of retrieving ?–B at a final test. This transfer effect has been established for word pairs [21]. However, when using word triplets the RP transfer effect is not seen, i.e. for A–B–C, RP for A–B–? does not transfer to B–C–? at final test [22]. Similarly, when the stimuli consist of more complex material such as prose passages, and more educationally relevant text such as concepts, facts and processes, the stimulus–response transfer effect is not observed. For example, RP for 'Thomas Jefferson purchased WHAT from France' does not transfer to 'Thomas Jefferson purchased Louisiana from WHOM?' apart from under specific RP conditions such as elaborate feedback methods [23,24].

The lack of transfer for A–B–C triplets and more complex prose passages is somewhat at odds with results suggesting that transfer effects can be seen for semantically related prose passages. For example, Chan et al. [19] found RP effects of transfer at a 24 h delay from 'Where do toucans sleep at night?' (answer: tree holes) to 'What other bird species is the toucan related to?' (answer: woodpeckers). In this example, the two questions are highly related because toucans use tree holes made by woodpeckers, a fact that was featured in the original study text. Here both the stimulus (question) and response (answer) are dissimilar; however, transfer effects are still present. Critically, Chan [17] went on to demonstrate that this transfer effect was dependent on the level of integration between material at encoding; when the material was presented in a coherent piece of prose (following a logical order) and participants were actively encouraged to integrate this material, transfer was seen at test 24 h later. However, when the sentence order of the prose passages was randomized, and no explicit instructions were given to integrate the information, accuracy for the related material was *lower* relative to a no RP condition. Thus, the level of integration at encoding can either facilitate or hinder RP transfer to related material.

The finding of a decreased RP effect for related material that is not actively integrated is in line with the 'retrieval-induced forgetting' effect [25]. Here, a category label (e.g. fruit) is paired with two category exemplars (e.g. apple and banana) at encoding. Following this, participants engage in repeated active

retrieval of one of the exemplars (e.g. apple) when cued with the category label. Retrieval accuracy is typically lower for the other exemplar (i.e. banana) relative to a non-tested category, suggesting that active retrieval of one exemplar subsequently impairs retrieval of the other related exemplar. Critically, this effect has been shown to decrease when participants are actively encouraged to integrate the two exemplars [26], again suggesting that the extent of integration can either facilitate or hinder RP transfer.

To summarize, while there is evidence for RP transfer effects, the conditions under which transfer occurs are not well understood. One clear boundary condition appears to be the extent to which the material is integrated at the point of encoding; however, other relevant factors include the delay between RP and final test [11], motivational factors and feedback complexity [23].

## 1.2. Pattern completion for integrated events

One explanation for the importance of integration in RP transfer is the concept of spreading activation [27,28]. Here, information is encoded in a network of associations that allows for the reactivation of related material via activation spreading from the cued information to associated information within the network [19,29]. The more highly associated the material, the more likely activity is to spread from the cued to the related material during RP, rendering transfer effects more likely.

A different, though related, concept is the process of pattern completion [30,31]. Here, the presence of a coherent representation is thought to allow for the retrieval of the complete representation (i.e. pattern) in the presence of a partial or ambiguous cue. This is similar to the concept of spreading activation in that non-cued, associated, information is retrieved; however, pattern completion is usually related to the retrieval of individual, coherent (episodic) memory traces [32,33], as opposed to spreading activation within a larger semantic network [28]. Crucially, however, both spreading activation and pattern completion accounts predict that RP should lead to transfer effects for well-integrated (but not poorly integrated) material.

Recent research has provided both behavioural [34] and fMRI [35,36] evidence for pattern completion in relation to so-called 'multielement events'. Here, participants learn to associate three distinct elements (e.g. a location, famous person and object) and at test are cued with a single element (e.g. location) and asked to retrieve one of the other elements (e.g. person). Behaviourally, the retrieval of elements within a specific event is statistically related—if you retrieve the location for that event successfully you are more likely to also retrieve the person and object for that event successfully (referred to as 'retrieval dependency' [37]). Further, fMRI evidence indicates that neocortical reinstatement of all event elements is evident—even for the task-irrelevant element for that trial (e.g. if cued with location and retrieving person, neocortical reinstatement also occurs for the related object [36]). This provides clear evidence for the integration and subsequent full re-activation of all associated elements (i.e. pattern completion). Thus, in this context RP transfer effects are likely to occur, given the strong evidence for coherent, integrated, mnemonic representations and knowledge of the underlying mechanisms that support the pattern completion process [38,39].

The present study is focused on examining RP transfer effects within multielement event triplets (in this case, locations, animals and objects). For example, if a participant actively retrieves the location–animal association during RP, does this enhance retrieval of the location–object and animal–object associations at final test due to pattern completion processes during retrieval? The reason for assessing RP transfer effects using this more 'episodic' paradigm is because of the strong empirical evidence for pattern completion. Further, the associated elements within a given triplet in this paradigm are semantically unrelated (cf. [19]). As such, any RP transfer effects must be due to the way in which the material is encoded, as opposed to being driven by potentially pre-existing semantic associations. If RP transfer effects are not seen in a paradigm such as this, where we know that integration is high and pattern completion occurs, then this places constraints on the likelihood of observing transfer effects in other experimental paradigms. Conversely, if transfer effects *are* seen, we will have an empirical basis for further investigation of the boundary conditions of transfer from tested to untested material, guided by the theoretical background associated with pattern completion.

## 1.3. Current study

We assessed RP transfer effects for multielement triplets, specifically testing for transfer from tested to untested associations and elements, within a given triplet. Participants learned a series of multielement triplets (locations, animals and objects; as in [40]). Each triplet was encoded under visual imagery conditions known to result in integration of the three elements [37]. Following

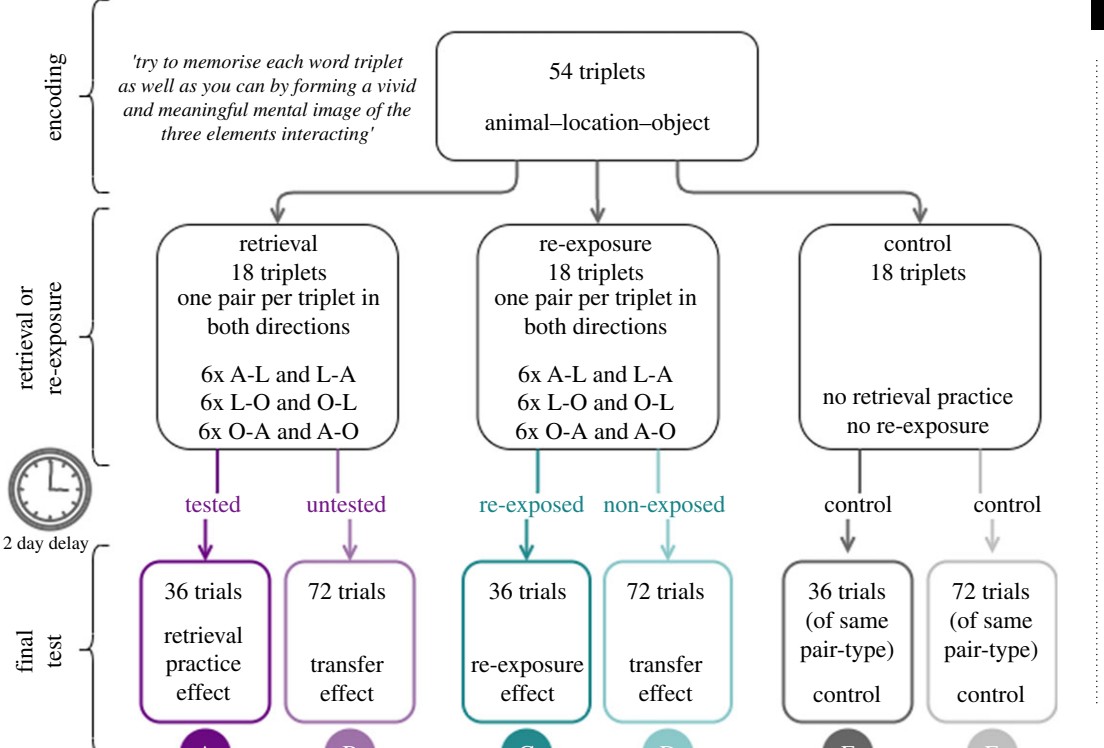

**Figure 1.** In the encoding phase, participants were presented with 54 word triplets (animal–location–object). During the retrieval practice and re-exposure conditions, participants saw one pair from 18 of the encoded event triplets in both directions (six *animal–location*, six *location–object* and six *object–animal*) for each condition. During the final test phase, participants were tested on every pair in both directions from every triplet studied during the initial encoding phase forming test conditions A–F for statistical analysis depending on whether they were tested/untested or re-exposed/nonexposed.

encoding, participants underwent RP for 1/3 of the triplets, and re-exposure for 1/3 of the triplets (the remaining 1/3 served as a nonexposed 'control'; figure 1).

The RP condition requires cued recall in response to a word cue and category cue (i.e. location, animal or object) followed by correct-answer feedback and the re-exposure condition provides participants with the word cue, category cue and the correct answer for re-study (figure 2). Importantly, for both the RP and re-exposure conditions, only one pairwise association per triplet was tested/re-exposed (although each association was tested twice in total; once in both directions over two separate blocks). For example, the location–animal association was tested, but not the location–object or animal–object association, which leaves the object element untested for that triplet. Following a 2-day delay (chosen to maximize the effects of RP; [11]), all pairwise associations for all triplets were tested with a four-alternative forced choice cued-recognition task (final test). Thus, for the RP and re-exposure conditions, we can assess memory performance for the directly tested/re-exposed pairs, as well as the untested/nonexposed pairs in the same triplets, allowing us to examine transfer effects.

A standard RP effect in this paradigm would manifest as higher accuracy at final test for the tested and re-exposed (through feedback) associations in the RP condition relative to the re-exposed (without retrieval) associations in the re-exposure condition. A transfer effect would present as higher performance for the untested/nonexposed associations in the RP condition relative to the nonexposed associations in the re-exposure condition. The non-re-exposed 'control' triplets provide a further means of assessing both the standard RP effect, as well as the transfer effect. To maximize the potential to see the RP effect, we incorporated the following methodological manipulations: (i) cued recall during RP, given evidence that recall relative to recognition produces greater RP and transfer effects [41,42], and (ii) a delay between RP and final test, given evidence that this maximizes both RP [11] and transfer [17].

First we aimed to replicate the robust finding that RP with feedback contributes to better performance at a delayed final test (see Hypothesis 1) compared to both the control condition (Hypothesis 1a) and re-exposure condition (Hypothesis 1b). Secondly, we aimed to investigate the existence of an RP transfer effect from tested elements to untested elements (Hypothesis 2) compared to the control condition (Hypothesis 2a) and re-exposure condition (Hypothesis 2b).

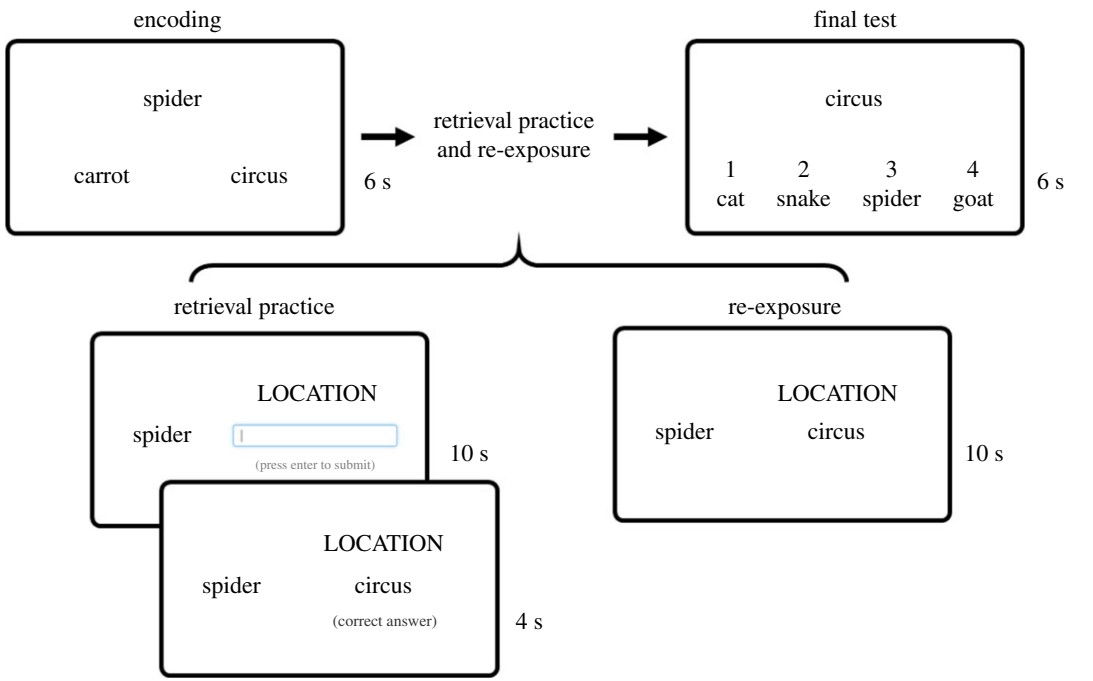

**Figure 2.** Trial types in each phase. In the encoding phase a word triplet was presented for 6000 ms in the format ANIMAL–LOCATION–OBJECT. In a Phase 2 retrieval practice trial participants were presented with the cue (e.g. spider) and a category cue (e.g. LOCATION) and had to type the location that they remember seeing paired with the cue within a 10 000 ms window. Next, participants received correct-answer feedback for 4000 ms. In a Phase 2 re-exposure trial participants saw the word cue, the category cue and the correct answer. In a Phase 3 final test trial participants saw the word cue and four multiple choice options with the corresponding keyboard number keys that they should press to select their answer within a 6000 ms window.

As discussed above, previous research has manipulated the stimulus–response arrangement between RP and final test with somewhat mixed results [24,43,44]. Here, we can directly assess the influence of repeating the stimulus or response in relation to RP transfer. If the location–animal association for a given triplet underwent RP, the location–object and animal–object associations will be the 'untested' pairs. Each of these associations was tested during final test in both directions. For example, on one trial, the location served as the cue (stimulus) and the object the target (response), whereas on another trial the object served as the cue and the location the target. In the former case, the stimulus (location) is repeated between RP and final test, but a different response is required (object at final test; animal during RP). In the latter case, the response (location) is repeated between RP and final test, but the stimulus changes (object at final test; animal during RP). By splitting these trial types, we can directly assess the extent to which RP transfer is driven by overlap in the cue (stimulus) or target (response) between RP and final test. Therefore, we compared accuracy on untested associations for the RP and re-exposure conditions where the cue is repeated but the required response is different (Hypothesis 3a), and where the cue is different but the required response is the same (Hypothesis 3b). Finally we compared trials where the cue is repeated but the required response is different with trials where the cue is different but the required response is the same for untested associations in the RP condition only (Hypothesis 3c).

To summarize, we conducted a pre-registered experiment using a paradigm known to induce integration of multiple semantically unrelated elements to test for RP transfer effects. We then assessed the extent to which these transfer effects are driven by repetition of the cue (stimulus) or target (response) between RP and final test. The study provides a strong empirical foundation for future research to investigate the boundary conditions of RP transfer, which has implications for best-practice application of RP in education settings.

## 2. Methods

### 2.1. Participants

Participants were recruited through Prolific (https://www.prolific.co/) in return for cash payment (up to £8 total; £4 upon completion of session one, and £4 upon completion of session two) with the following

pre-screening rules applied via Prolific: participants must have been using a laptop or desktop PC, be aged 18–35, and be native English speakers. On Gorilla (https://gorilla.sc/) [45]), where the study was hosted, additional screening ensured that participants were using a laptop and desktop PC, and using either Chrome, Firefox, Edge, Internet Explorer or Safari web browsers. The consent form and instructions asked that participants confirm that they either had normal vision or corrected-to-normal vision. We continued to recruit participants until we had a suitable number of usable datasets (see *Data collection stopping rules* section) defined as those which remain in the sample after applying the criteria outlined in the *Data exclusion* section. All participants provided informed consent prior to participating. The study was approved by the Department of Psychology's research ethics committee at the University of York (ref. 875).

## 2.2. Data collection stopping rules

Our main effect of interest was a paired-samples one-sided *t*-test comparing untested pairs from RP triplets to unstudied pairs from re-exposed triplets (see Hypothesis 2b). We ran a similar small-scale (unpublished) study that investigated transfer effects using the same paradigm as here, albeit compared to a control condition rather than a re-exposure condition, and found a significant transfer effect with an effect size of $d = 0.51$. According to a recent meta-analysis the 95% lower bound for RP transfer effects including re-exposure is $d = 0.31$ [13], and so we used a more conservative estimate of the effect size. Using the *pwr* package in RStudio [46,47] we performed a power analysis for the *t*-test of interest using this lower-bound estimate ($d = 0.31$), an alpha level of 0.025 (to account for family-wise error), and with a power of 0.9 (power analysis scripts: https://osf.io/wtyku/). The resulting estimate was $n = 112$. Resource constraints allowed testing of 150 participants maximum, so we pre-registered that we would continue data collection until we reached 112 usable datasets or 150 datasets in total (whichever we reached first). In the instance where we could have 150 datasets and less than 112 usable datasets, we were likely to still have at least 0.8 power which, using the same method in R, required 84 participants/usable datasets. The attrition rate from past online studies from our group, using similar experimental designs with a longitudinal element, has ranged from 10% to 20% (participants lost to data exclusion criteria such as attention checks and performance accuracy, as well as drop-out rates between two sessions), which suggested that we would comfortably reach at least 0.8 power, presuming a transfer effect is the same magnitude or larger than the 95% lower bound for RP transfer effects estimated in a systematic meta-analysis [13].

## 2.3. Stimuli

The stimuli were 54 word triplets each consisting of an animal, an object and a location. As in James *et al*. [40], animal characters were used instead of famous people (e.g. [37]) to make the task accessible to a wider range of age groups in future studies. The triplets were split into three stimulus sets and were counterbalanced across the three conditions. The lists of animals, objects and locations within each set have been rated and matched for age of acquisition [48], imageability [49–51], number of syllables per word and concreteness [52]. Stimuli and related information are available (https://osf.io/wtyku/).

## 2.4. Procedure

The study took place on the online Gorilla platform [45] and participants were recruited via Prolific. The experiment is available through Gorilla Open Materials (https://app.gorilla.sc/openmaterials/107080). Participants were able to contact the researcher via Prolific's messaging system at any point during their participation. Participants were shown the information sheet, prompted to fill in the consent form to continue, and asked to provide their date of birth and gender with the following options: *male*, *female*, *prefer not to say*, *prefer to self-describe (please specify below)*. If the latter option was chosen, a free-text box became available. Participants were asked to take part in two sessions separated by 2 days; in session one they completed an encoding phase and a retrieval/re-exposure phase, and in session two they completed a final test phase.

The opportunity to participate in session one went live on Prolific between 9.00 and 10.00 BST, with a 12-h time limit to ensure that participants finished the first session that same day. Invitations for session two were then sent to participants who completed session one 2 days later between 9.00 and 10.00 BST, with the expectation that they would finish the task that same day. This allowed us to achieve a balance between controlling the delay between the sessions while allowing the participants some flexibility in

their time zone. We made it clear in session one that they should expect the invitation for session two in 2 days' time and encouraged them to make some time in their schedule to complete the session.

### 2.4.1. Phase 1: encoding

The encoding phase was split into three blocks of 18 trials each for a total of 54 trials, and trial order was randomized for each participant. Before the encoding phase, participants were instructed to 'try to memorize each word triplet as well as you can by forming a vivid and meaningful mental image of the three elements interacting' and then given the opportunity to perform two practice trials. Each trial consisted of a three-element word triplet consisting of an animal, a location and an everyday object (e.g. *spider–circus–carrot*) presented on screen for 6000 ms followed by a blank 500 ms inter-trial interval.

The animal, location and object were presented in a triangle formation in the centre of the screen (figure 2). Note that the precise locations, word size and visual angle between words varied slightly dependent on the participants' screen size, resolution and other display settings.

There was one attention check per block during the encoding phase to make sure participants were paying attention, which participants were warned of during the block instructions. Randomly during the block, a screen appeared that said 'Attention check: PRESS THE SPACEBAR!' and participants had 5 s to follow this instruction (a countdown was displayed on screen). Participants who failed any of the three attention checks were not able to proceed with the second session and were screened out of the study (see section on *Data exclusion*).

### 2.4.2. Phase 2: retrieval practice or re-exposure

Immediately after the encoding phase participants completed blocks of RP and re-exposure trials. During the RP condition, participants were presented with a cue containing one element from one of the triplets that they saw in the encoding phase and given a category cue (i.e. animal, location, object). Participants had to type in the element that belongs to that category that they remembered being paired with the cue during the encoding phase (figure 2). For example, for *spider–circus–carrot* they saw *circus* as the element cue, *animal* as the category cue, and using cued recall typed in the *animal* that they saw paired with *circus* (i.e. *spider*). Within a 10 000 ms timeframe, participants were asked to type in their answer and press the enter key when they were ready to submit it; for the last 3 s a countdown appeared on screen to signal the time remaining. In the instructions, they were encouraged to make their best guess if they did not know the answer. If participants did not respond, they were prompted with a reminder to guess before the next trial began. Once they had submitted their answer participants received correct-answer feedback regardless of the accuracy of their own answer. Their typed answer was replaced with the correct answer for 4000 ms.

During the re-exposure condition participants saw the correct-answer feedback screen only, which was displayed for 10 000 ms in order to keep the overall trial length as similar to possible to an RP trial (which was variable, dependent on how quickly participants typed their cued recall answer). Participants were instructed to 'try and commit those word pairs to memory. You'll be given 10 s in which to try and memorize the pair before it moves on to the next one' to try and prevent any active retrieval. As in the encoding phase, we included one attention check per block for the re-exposure condition only. Participants who failed any attention check were not invited back to complete the second session.

Participants were tested on one pairwise association from 18 of the 54 encoded triplets in the RP condition, in both cue–target directions (six cue *animal*–retrieve *object* and vice versa, six cue *animal*–retrieve *location* and vice versa and six cue *object*–retrieve *location* and vice versa), and one pairwise association from 18 triplets in the re-exposure condition (six of each cue and retrieval type as before). Therefore, each tested/re-exposed association was seen twice in total during this phase. Although there is evidence to suggest that the testing effect may increase with multiple RP trials (e.g. [53–55]), the effect has still been established to be robust with only a single trial [6]. The remaining 18 triplets acted as the control triplets and were not included in the RP or re-exposure conditions. Of the RP/ re-exposure triplets, only one pair of elements was tested/re-exposed in both directions (e.g. *animal–location* and *location–animal*) and the remaining two pairs were untested (see 'Retrieval or re-exposure' in figure 1), and so there were a total of 36 trials per condition.

Participants first completed one block containing half (18) of the trials for condition A (retrieval or re-exposure, depending on counterbalancing), a second block containing half of the trials for condition B (retrieval or re-exposure, depending on counterbalancing), then two more blocks containing the

remaining trials for condition A and then condition B, respectively. We opted for a blocked design rather than randomizing the conditions trial by trial to reduce effects of task-switching, and to reduce any difficulty for participants in comprehending task instructions in the online environment where they are less likely to ask questions of the researchers for clarity. Previous laboratory-based research suggests RP is robust in both a mixed and blocked design [21].

Participants were assigned to a counterbalancing order automatically by Gorilla upon completing the consent form with 18 possible assignments to control for the following: (i) stimulus sets 1–3 counterbalanced across RP, re-exposure and control conditions, (ii) the untested element within each triplet from each set (three sets) and (iii) whether they completed the RP or re-exposure condition first in an ABAB design. After completion of this phase, participants were prompted to complete an exit questionnaire (see section on *Exit questionnaire*), reminded that they would receive an invitation through Prolific in 2 days' time, and received payment for session one.

### 2.4.3. Phase 3: final test

In the second session, 2 days later, participants that passed the data quality checks for session one (see section on *Data exclusion*) completed a final multiple-choice memory test which was sent to them through Prolific. Every pairwise combination of elements within each of the 54 triplets that the participant learned in the encoding phase was tested in both directions.

Participants were presented with a cue which was one element (animal, object or location) from one of the triplets in the encoding phase, and provided with four elements to choose from that all belonged to one of the two possible categories (e.g. cue *animal* and retrieve *object*). One of the four elements was associated with the cue at encoding, and the other three elements (foils) were randomly selected from any of the remaining 53 triplets (i.e. regardless of the condition at Phase 2). Using the one to four number keys along the top of their keyboard participants had to select which option that they remembered being paired with the cue during the encoding phase or to make their best guess. Participants had 6000 ms to select an answer, and the trial moved on to a blank inter-trial interval of 500 ms either when an answer had been selected or the trial timed out, whichever occurred first. Participants were encouraged to respond on every trial. If no response was given, these trials were classified as incorrect. No feedback was given in the final test phase. The test phase was split into six blocks of 54 trials for a total of 324 trials. Multiple-choice final tests have been shown to produce medium-to-large effect sizes in a recent meta-analysis [6].

### 2.4.4. Exit questionnaire

After completing session one, participants were provided with an exit questionnaire consisting of four questions to aid in assessing data quality when running unsupervised studies online: (1) Please briefly describe any strategies you used to learn the animals, objects and locations, (2) Did anyone else help you with this task? If so, please describe, (3) Did you use any memory aids? (e.g. writing things down, any other strategies) and (4) Is there anything else that might be helpful for us to know? (e.g. technical issues, etc.). Question (1) allowed us to assess how well people adhered to the instructions to visualize each word triplet interacting in a meaningful way and was used to inform future studies. Questions (2)–(4) served as quality control checks which are detailed in the *Data exclusion* section. After completing session two, participants were asked question (4) only for a quality control check (see *Data exclusion* section).

# 3. Analysis

## 3.1. Data processing

Accuracy for RP trials was first rated automatically, and any remaining trials manually rated by the researchers. Using RStudio, participants' responses were checked to see if they matched the expected response (a correct response) or if the trial timed out without a response (a missed response). Next, typed responses were checked against a custom dictionary of likely typographical errors created by the researchers which included potential misspellings such as spaces removed/added, incomplete word stems, double letters, missed letters, etc. If the participants' response matched an entry in the dictionary of accepted typographical errors, the error was corrected in the dataset.

Next, all uncategorized responses were checked again to see if they matched the expected response (a correct response), if the response was a within-triplet category error (e.g. the participant was cued with an animal word, asked for LOCATION but instead provided the correct OBJECT), a within-category triplet error (the participant gave a response from the correct category but which was featured in another triplet from the stimulus list), or a between-triplet category error (the participant gave a response that was featured elsewhere in the stimulus list but was not from the focal triplet nor from the focal category), to further aid classification of correct or incorrect responses. Any remaining trials that could not be automated by R were manually (and independently) rated by two researchers. In the event of disagreement, a third researcher decided on the classification.

Although participants were told to press enter upon finishing their answer, any text that was in the response box was recorded by Gorilla regardless of whether they pressed enter or not. In the event of incomplete word stems that contained the first two or more matching characters (e.g. 'fro' instead of 'frog'), these were scored as correct. Answers were also considered correct if they were a typographical error (e.g. 'forg' instead of 'frog'), but not if they were semantically related but incorrect (e.g. 'toad' instead of 'frog').

The exit questionnaires were screened by one researcher to identify participants that should potentially be excluded (see *Data exclusion* section). Another researcher inspected these cases and, if in agreement, their data were excluded.

## 3.2. Data exclusion

Participants' data were excluded and they were not invited back to participate in session two if, during session one, they provided an age outside of our requested inclusion criteria (18–35), they failed any of attention checks across encoding and re-exposure trials, they achieved an accuracy of less than 20% during the cued recall RP trials (including trials where they did not provide an answer) after the automated process of identifying typographical errors and detecting errors but before the researchers manually checked the remaining responses, or they reported using a memory aid/help from another person/significant technical issues during the exit questionnaire (see section on *Data processing* for managing qualitative data exclusion criteria).

Participants who were eligible to proceed to session two were excluded from analysis if they had not returned to complete the second session within 24 h of the study going live, achieved an accuracy of less than 30% or greater than 95% (collapsed across conditions) at final test, or reported significant technical issues (see section on *Data processing* for managing qualitative data exclusion criteria).

## 3.3. Hypotheses and statistical analyses

Details of the pre-registered hypotheses and their corresponding statistical analyses and possible interpretations can be found in the Stage 1 Registered Report (https://osf.io/qgah7). Pre-registered analyses are clearly separated from exploratory analyses throughout this paper. Figure 1 shows how each condition at final test maps onto a measure of accuracy. All statistical tests are within-subject *t*-tests (either one-tailed or two-tailed, dependent on the hypothesis). Alongside *t*-statistics and Cohen's *d* effect sizes (mean difference between the conditions divided by the pooled standard deviation across conditions as an estimate of the between-subjects effect size), we also report Bayes factors to complement the main null hypothesis significant testing approach. Bayes factors were computed using the *BayesFactor* package in R [56] and using a default prior Cauchy distribution of $r = 0.707$ centred at 0. Where the Bayes factors indicate that we do not have enough evidence to support our findings (i.e. a Bayes factor between 0.33 and 1 [57]), we discuss the null hypothesis significance tests in the appropriate context.

### 3.3.1. Retrieval practice effect (Hypothesis 1)

The RP effect is robust in the literature, and so first we aimed to conceptually replicate previous findings and demonstrate that accuracy for each pair at final test changes as a function of RP.

If the RP effect has occurred in our study, we would expect accuracy for associations tested with RP to be higher than control trials (Hypothesis 1a). To test this, we performed a one-tailed *t*-test on the difference in accuracy between the RP associations (test condition A) and the equivalent control associations (test condition E). We expected accuracy to be significantly higher for RP trials relative to controls.

The RP effect is shown to be a robust effect that improves retention over re-exposure (and no retrieval). We should, therefore, see greater accuracy for tested pairs from RP triplets relative to re-exposed pairs for re-exposed triplets (Hypothesis 1b). We performed a one-tailed *t*-test on the accuracy at final test between the tested pairs from RP triplets (test condition A) and the re-exposed pairs from re-exposed triplets (test condition C). We expected accuracy in the RP condition to be significantly higher than in the re-exposure condition.

### 3.3.2. Retrieval practice transfer effect (Hypothesis 2)

If transfer occurs from tested to untested material, we would expect that the untested pairs from RP triplets would show higher accuracy at final test compared to control and re-exposed triplets.

As in Hypothesis 1a, we first assessed transfer relative to the 'control' condition, comparing untested pairs from RP triplets to control triplets (Hypothesis 2a). We performed a one-tailed *t*-test on accuracy of untested pairs from RP triplets (test condition B) to control triplets (test condition F). We expected accuracy to be significantly higher in the untested RP trials compared to the control trials.

We next compared untested pairs from RP triplets to nonexposed pairs from re-exposed triplets, assessing whether transfer is specifically related to RP relative to re-exposure (Hypothesis 2b). We performed a one-tailed *t*-test on the accuracy of the untested pairs from RP triplets (test condition B) to nonexposed associations from re-exposed triplets (test condition D). We expected RP to enhance any transfer effects and thus for accuracy to be significantly higher for trials of untested pairs from RP triplets compared to trials of nonexposed pairs from re-exposed triplets.

### 3.3.3. Transfer as a function of stimulus–response congruency (Hypothesis 3)

We performed three planned comparisons to assess the extent to which a transfer effect, if any, is driven by stimulus or response repetition between RP and final test.

We performed a one-tailed *t*-test on the accuracy at final test of the untested pairs from RP triplets where the cue is the same as during RP but the target is different, compared to the nonexposed pairs from re-exposed triplets for the same *repeat cue–different target* trials (Hypothesis 3a). We expected to find a transfer effect had occurred in the RP compared to the re-exposure condition.

We performed a one-tailed *t*-test on the accuracy at final test of the untested pairs from RP triplets where the target is the same as during RP but the cue is different, compared to the nonexposed pairs from re-exposed triplets for the same *different cue–repeat target* trials (Hypothesis 3b). We expected to find a transfer effect had occurred in the RP compared to the re-exposure condition.

Although both *repeat cue–different target* and *different cue–repeat target* trials may contribute to RP transfer effects (dependent on the results of Hypotheses 3a and 3b), they may not equally contribute to transfer. To test this, we performed a two-tailed *t*-test on accuracy for *repeat cue–different target* compared to *different cue–repeat target* for the RP condition only (Hypothesis 3c). We had no directional hypothesis in relation to whether repeating the cue or target will produce greater transfer (hence the two-tailed *t*-test).

# 4. Results

## 4.1. Participants

The final dataset consisted of 113 participants with a mean age of 26.56 (s.d. = 4.96, 18–35). Sixty-five identified as female, two as non-binary, one as a transgender male and 45 as male. Of the initial 346 participants that provided informed consent on Gorilla for Session 1, 21 left the study before finishing, 25 failed the attention checks in the encoding phase, 11 failed the attention checks in the re-exposure phase, 22 were removed due to technical issues with some of the stimuli on the first 2 days of testing (this was resolved for the remainder of the testing period), 128 were removed for low accuracy during cued recall (less than 20%), 2 provided an age outside of the 18–35 criteria and 1 was excluded in the exit questionnaire. We discuss the high exclusion rate for low accuracy during RP in the discussion. Of the remaining 136 participants that were eligible to continue with Session 2, 124 returned. One participant was excluded for experiencing technical issues during their participation, and 10 for low accuracy (less than 30%) at final test. As we overrecruited on each day of data collection to allow for participant attrition (due to not returning to complete the task or being excluded for high or low accuracy), this meant that on the final day of data collection we achieved 113 usable datasets and,

**Table 1.** Means and standard deviations for percentage of error types during the cued recall task in the retrieval practice phase.

| response accuracy | mean (%) | s.d. (%) |
|---|---|---|
| correct | 43.39 | 16.03 |
| error: no response provided | 4.79 | 9.08 |
| error: wrong event, correct category | 37.00 | 15.84 |
| error: correct event, wrong category | 1.92 | 2.85 |
| error: wrong event, wrong category | 3.56 | 3.47 |
| error: response was not an item from the stimulus set | 9.34 | 7.25 |

**Table 2.** Mean accuracy (and standard deviations) for all test conditions related to the retrieval practice hypotheses, the transfer hypotheses and the stimulus–response congruency hypotheses.

| condition | mean accuracy (%) | s.d. (%) |
|---|---|---|
| retrieval practice and transfer | | |
| tested RP associations (test condition A) | 69.20 | 16.94 |
| untested RP associations (test condition B) | 52.85 | 16.70 |
| re-exposed re-exposure associations (test condition C) | 67.38 | 18.30 |
| nonexposed re-exposure associations (test condition D) | 54.82 | 17.63 |
| equivalent control pairs for test conditions A and C (test condition E) | 42.87 | 14.88 |
| equivalent control pairs for test conditions B and D (test condition F) | 43.94 | 14.38 |
| stimulus–response congruency | | |
| RP: same cue, same target (i.e. test condition A) | 69.20 | 16.94 |
| RP: same cue, different target (repeat cue) | 54.42 | 17.81 |
| RP: different cue, same target (repeat target) | 51.28 | 16.94 |
| re-exposure: same cue, same target (i.e. test condition C) | 67.38 | 18.30 |
| re-exposure: same cue, different target (repeat cue) | 55.63 | 17.92 |
| re-exposure: different cue, same target (repeat target) | 54.01 | 18.65 |

as we had not yet examined the results of the data, we elected to include all participants over our threshold of 112 usable datasets.

## 4.2. Accuracy for retrieval practice trials

In the RP phase participants were, on average, correct on 43.39% of trials (s.d. = 16.03%). A substantial portion of errors (37 ± 15.84%) were due to participants providing a response that was from the correct category but from a different event within the stimulus set. Reassuringly, trials where participants incorrectly provided the element that, for experimental purposes, was not intended to undergo active retrieval (i.e. their response was the untested element; from the correct event, but the wrong category) were low (1.92% ± 2.85%) which allows us to separate RP from transfer effects confidently throughout the results. Full details of mean error types are in table 1.

## 4.3. Main analyses

Descriptive data are presented in table 2 and statistical tests in table 3. All analysis was conducted with R in RStudio [47] using the *BayesFactor* [56], *broom* [58], *cowplot* [59], *janitor* [60], *lubridate* [61] and *tidyverse* [62] packages.

**Table 3.** Statistical results from the numbered pre-registered hypotheses as well as additional exploratory analysis.

| | test statistic | p-value | 95% CI | Cohen's d | $BF_{10}$ |
|---|---|---|---|---|---|
| **retrieval practice effect** | | | | | |
| *Hypothesis 1a.* Accuracy is significantly higher for tested pairs from RP triplets than for equivalent pairs from control triplets | t = 17.04 (d.f. = 112) | <0.001[a] | [0.23, inf][b] | 1.65 | $4.25 \times 10^{29}$ |
| *Hypothesis 1b.* Accuracy is significantly higher for tested pairs from RP triplets than for equivalent pairs from re-exposure triplets | t = 1.48 (d.f. = 112) | 0.07 | [−0.002, inf] | 0.10 | 0.30 |
| *Exploratory analysis 1c.* Accuracy is significantly higher for re-exposed RP triplets than for equivalent pairs from control triplets | t = 16.03 (d.f. = 112) | <0.001[a] | [0.22, inf] | 1.47 | $3.39 \times 10^{27}$ |
| **retrieval practice transfer effect** | | | | | |
| *Hypothesis 2a.* Accuracy is significantly higher for untested pairs from RP triplets than for equivalent pairs from control triplets | t = 8.72 (d.f. = 112) | <0.001[a] | [0.07, inf] | 0.57 | $2.37 \times 10^{11}$ |
| *Hypothesis 2b.* Accuracy is significantly higher for untested pairs from RP triplets than for equivalent pairs from re-exposure triplets | t = −1.63 (d.f. = 112) | 0.94 | [−0.04, inf] | −0.11 | 0.38 |
| *Exploratory analysis 2c.* Accuracy is significantly higher for nonexposed pairs from re-exposure triplets than for the equivalent pairs from control triplets | t = 9.48 (d.f. = 112) | <0.001[a] | [0.09, inf] | 0.68 | $1.16 \times 10^{13}$ |
| **transfer as a function of stimulus–response congruency** | | | | | |
| *Hypothesis 3a.* Accuracy where the cue is repeated but the target is different is higher for untested pairs from RP triplets than for nonexposed pairs from re-exposure triplets | t = −0.90 (d.f. = 112) | 0.90 | [−0.03, inf] | −0.07 | 0.15 |
| *Hypothesis 3b.* Accuracy where the cue is different but the target is the same is higher for untested pairs from RP triplets than for nonexposed pairs from re-exposure triplets | t = −2.01 (d.f. = 112) | 0.98 | [−0.05, inf] | −0.15 | 0.72 |
| *Hypothesis 3c.* Accuracy for untested pairs from RP triplets is different depending on whether the cue is repeated, or the target is repeated | t = 3.46 (d.f. = 112) | <0.001[a] | [0.01, 0.05] | 0.18 | 27.18 |
| *Exploratory analysis 3d.* Accuracy for nonexposed pairs from re-exposure triplets is different depending on whether the cue is repeated, or the target is repeated | t = 1.78 (d.f. = 112) | 0.08 | [−0.001, 0.03] | 0.09 | 0.48 |

[a]Denotes statistical significance at the pre-registered alpha level or, in the case of exploratory tests, an adjusted alpha level to account for the (new) total number of statistical tests within that family (see main text for details). All t-tests are one-tailed except analyses 3c and 3d which are two-tailed.

[b]Upper bound is infinite due to the nature of one-tailed tests.

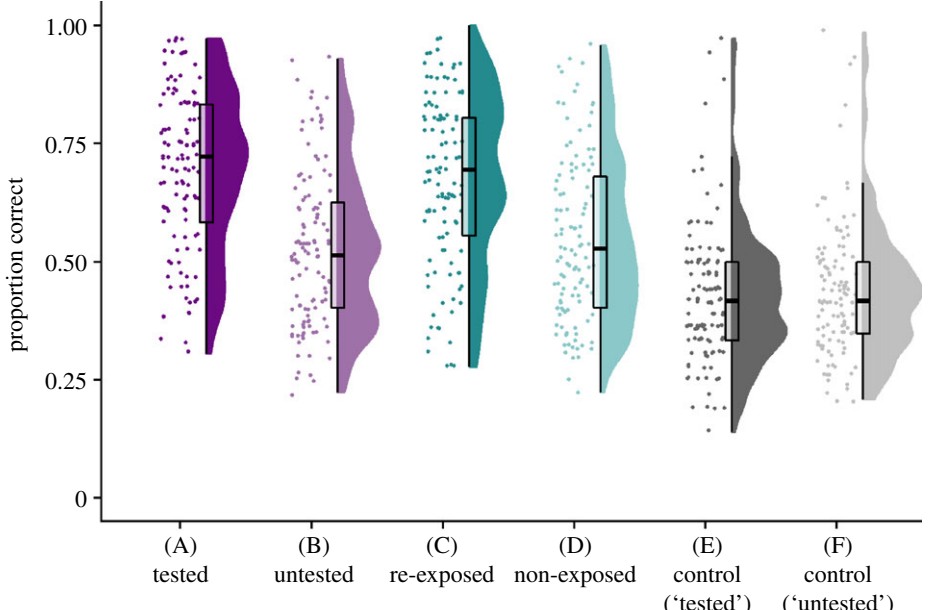

**Figure 3.** Raincloud plots [63] show each participant's raw data (horizontally jittered), a boxplot and split half violin of the density for each pair-type at final test. Further information on test conditions can be found in figure 1.

### 4.3.1. Retrieval practice effect (Hypothesis 1)

Data for the RP effect are shown in figure 3 where the relevant conditions are A, C and E. As predicted, accuracy for the tested RP associations (test condition A) was significantly higher than that for the equivalent control trials (test condition E), $t_{112} = 17.04$, $p < 0.001$, $BF_{10} = 4.25 \times 10^{29}$, $d = 1.65$.

However, contrary to our predictions, there was no significant difference between the tested RP associations (test condition A) and the re-exposure pairs from the re-exposed triplets (test condition C), $t_{112} = 1.48$, $p = 0.07$, $BF_{10} = 0.30$, $d = 0.10$. The BF suggests that we may not have enough evidence in this sample, although there is more evidence for the null hypothesis than the alternative.

To examine the RP effect further, we conducted an exploratory $t$-test to see if accuracy for associations that were re-exposed was higher than the equivalent control trials. Accuracy for re-exposed pairs from re-exposure triplets (test condition C) was significantly higher than the equivalent control trials (test condition E) using a one-tailed $t$-test with an alpha level of 0.016 (to account for this being the third test in this family), $t_{112} = 16.03$, $p < 0.001$, $BF_{10} = 3.39 \times 10^{27}$, $d = 1.47$. In sum, we saw greater accuracy for both the tested RP and re-exposure pairs from the re-exposed triplets relative to control pairs; however, no difference was seen between tested RP pairs and re-exposure pairs.

### 4.3.2. Retrieval practice transfer effect (Hypothesis 2)

Data for the transfer effect are shown in figure 3 where the relevant conditions are B, D and F. As predicted, accuracy for untested pairs from RP triplets (test condition B) was significantly higher than that for the equivalent pairs from control triplets (test condition F), $t_{112} = 8.72$, $p < 0.001$, $BF_{10} = 2.37 \times 10^{11}$, $d = 0.57$.

Assessing whether transfer was specifically related to RP relative to re-exposure, accuracy of the untested pairs from RP triplets (test condition B) was not significantly higher than accuracy for the nonexposed pairs from re-exposed triplets (test condition D), $t_{112} = -1.63$, $p = 0.94$, $BF_{10} = 0.38$, $d = -0.11$.

Finally, to examine transfer effects further, we conducted an additional exploratory $t$-test to see if accuracy was higher for nonexposed pairs from re-exposure triplets (test condition D) compared to equivalent control trials (test condition F). A one-tailed $t$-test with an alpha level of 0.016 (to account for this being the third test in the family) showed that accuracy was significantly higher for nonexposed pairs compared to control pairs, $t_{112} = 9.48$, $p < 0.001$, $BF_{10} = 1.16 \times 10^{13}$, $d = 0.68$. We, therefore, saw evidence of transfer when comparing both the untested pairs from RP triplets and nonexposed pairs from re-exposure triplets to control pairs; however, no difference in accuracy was seen between the untested/nonexposed pairs RP triplets relative to re-exposed triplets.

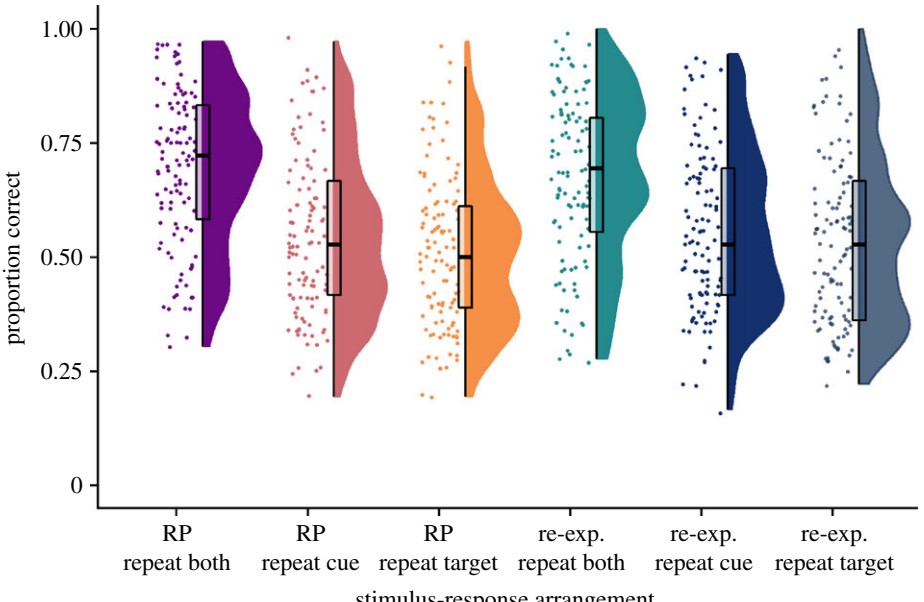

**Figure 4.** Raincloud plots show each participant's raw data (horizontally jittered), a boxplot and split half violin of the density for each stimulus–response arrangement pair-type at final test. 'RP' refers to the retrieval practice pairs and 're-exp.' refers to the re-exposure pairs. The 'RP repeat both' and 're-exp. repeat both' conditions are equivalent to test conditions A and C, respectively, as shown also in figure 3.

### 4.3.3. Transfer as a function of stimulus–response congruency (Hypothesis 3)

Data for the transfer effect as a function of stimulus–response congruency are shown in figure 4, where the first and fourth rainclouds show the general RP and re-exposure effects respectively, where both the cue and target are repeated from the Phase 2 trials (i.e. rainclouds A and C from figure 3).

Accuracy was not significantly higher for *repeat cue–different target* trials in the RP condition (raincloud two, figure 4) compared to the re-exposure condition (raincloud five, figure 4), $t_{112} = -0.90$, $p = 0.90$, $BF_{10} = 0.15$, $d = -0.07$, and the BF was inconclusive. For *different cue–repeat target* trials (rainclouds three and six, figure 4), accuracy was not significantly higher in the RP condition compared to the re-exposure condition, $t_{112} = -2.01$, $p = 0.98$, $BF_{10} = 0.72$, $d = -0.15$.

Although we found no evidence for differential transfer effects for either *repeat cue–different target* or *different cue–repeat target* trials between RP and re-exposure conditions, they may contribute differentially within the RP condition alone. We found higher accuracy for *repeat cue–different target* trials than for *different cue–repeat target* trials, $t_{112} = 3.46$, $p < 0.001$, $BF_{10} = 27.18$, $d = 0.18$, suggesting that repetition of the cue improves retrieval relative to repetition of the target.

Finally, as we found that transfer effects in the RP condition differed depending on whether the cue or target was repeated between RP and final test (Hypothesis 3c), and we had found no evidence of an RP effect or transfer effect over and above re-exposure (Hypotheses 1b and 2b, respectively), we performed an additional exploratory analysis to examine stimulus–response arrangement in the re-exposure condition only. A two-tailed *t*-test (alpha level = 0.0125 to account for this being the fourth test in the family) showed that accuracy for *repeat cue–different target* trials was not significantly different from *different cue–repeat target* trials in the re-exposure condition ($t_{112} = 1.78$, $p = 0.08$, $BF_{10} = 0.48$, $d = 0.09$). We, therefore, found tentative evidence for greater RP transfer effects when the cue is repeated relative to when the target is repeated, a pattern that was not seen for re-exposure triplets.

### 4.4. Additional exploratory analyses

Participating in the RP condition first may have elicited automatic retrieval in the subsequent re-exposure session. As we found no RP effect, we performed additional exploratory analyses to check that there was no difference in final test performance between those participants that did RP before re-exposure, compared to those that did re-exposure before RP. In a two-way mixed ANOVA with a between-subjects factor of group (whether the participants did the RP condition first, or the re-exposure condition first) and a within-subjects factor of pair-type at final test (tested pairs from the RP triplets

versus re-exposed pairs from the re-exposure triplets), we examined accuracy at final test. The ANOVA showed no significant main effect of group ($F_{1,222} = 0.40$, $p = 0.53$) or pair-type at final test ($F_{1,222} = 0.60$, $p = 0.44$) nor a significant interaction ($F_{1,222} = 1.54$, $p = 0.22$). We, therefore, found no evidence that the order of RP and re-exposure blocks affected the pattern of accuracy results at final test.

Finally, after data collection, we found an error in the Final Test phase of the experiment where four of the events contained one incorrect element that was different from the events seen at encoding and during retrieval/re-exposure. This affected 2 (out of 6) retrieval trials per event for 4 (out of 54) events (8 out of a total of 324 trials). Importantly, these events were rotated across conditions between participants. We included all trials and events in the main analyses above, but as an exploratory measure we checked to see if the results changed if we removed the four events from the dataset. The significance values of all statistical tests remained the same and no results or conclusions were altered by this error.

# 5. Discussion

Despite robust evidence for the benefits of RP for the retention of directly tested information [3,6], whether these benefits transfer to associated but non-tested material is less clear [13]. In this registered report, we used event triplets to examine the extent to which RP on tested pairs transferred and led to better retention for untested pairs from the same event triplet. We found evidence for transfer, where memory performance was higher for untested pairs from RP triplets than pairs from control triplets that were not retrieved. However, we also observed a similar transfer effect for nonexposed pairs from re-exposed triplets relative to control triplets. Thus, we provide evidence for transfer (relative to a low-level control condition) for triplets that underwent either RP or re-exposure. Importantly, we also saw higher memory performance for both directly retrieved and re-exposed pairs relative to the control condition and did not see differences between the RP and re-exposure conditions. Our results demonstrate that transfer can occur for event elements that are not directly tested or re-exposed, but only if associated event elements have been tested or re-exposed. These findings question the robustness and/or boundary conditions of RP effects relative to simple re-exposure. We discuss these findings in turn.

## 5.1. Transfer effects

We used event triplets, consisting of a location, object and animal, presenting all three elements in a single encoding trial, and encouraging participants to engage in mental imagery where the three elements interacted. This design was used to encourage the integration of the three semantically unrelated elements. Previous research has shown that the retrieval of such triplets is underpinned by a hippocampal pattern completion process, leading to the retrieval of all elements (even when not task-relevant for that trial) [34–37]. We reasoned that such conditions would encourage transfer effects from tested to untested elements within a triplet, providing an experimental approach to explore the boundary conditions of RP transfer. Conversely, if RP transfer was not seen despite the highly integrated nature of the triplet elements, this would suggest that transfer is unlikely to occur in other settings. The presence of transfer (relative to the not repeated control condition) suggests that memory performance can be boosted for untested material if it is directly associated with the tested material (where either the cue or target is repeated across tested and untested pairs). Given the prior theoretical work relating the retrieval of integrated triplets to the computational process of pattern completion, we believe these transfer effects are most likely driven by the incidental retrieval of all triplet elements during RP trials.

Interestingly, we saw similar evidence of transfer effects for triplets that underwent re-exposure relative to the not repeated control condition. Namely, memory performance was higher for nonexposed pairs in re-exposed triplets (relative to the control condition), suggesting the re-exposure of individual pairs is sufficient to increase retention for material that is directly associated with the re-exposed material. Given a lack of evidence for any differences between the RP and re-exposure conditions, the most parsimonious explanation for transfer within re-exposure triplets is the incidental retrieval of all triplet elements during re-exposure trials (as is likely the case for the RP trials). The finding of transfer during both RP and re-exposure is pedagogically important, as it suggests under certain conditions, such as when event information is initially presented in a highly integrated manner, transfer can be induced via repetition, with or without effortful retrieval.

## 5.2. Retrieval practice versus re-exposure

By contrast to many studies of RP [6], we did not see a difference in memory performance between the RP and re-exposure condition, either for retrieved versus re-exposed pairs or not retrieved versus nonexposed pairs (from retrieved/re-exposed triplets). This questions the ubiquity and robustness of RP in relation to boosting retention relative to simple re-exposure. We specifically designed our study, based on prior literature, to increase the chances of demonstrating a robust RP effect (see Introduction and Methods for detail). Our findings, therefore, suggest the field has not yet fully explored the boundary conditions of RP, in relation to the experimental design and learning material.

Interestingly, studies on retrieval-induced forgetting (RIF) have also shown similar memory performance following retrieval versus re-exposure. Here, the repeated retrieval of an item when presented with an associated cue can cause forgetting for a separate item that was also associated with the same cue (in an A–B, A–C design) [25]. Forgetting of associated items is only seen following active retrieval, and not simple re-exposure [64–66]. Critically, however, facilitation for the retrieved/ re-exposed items (relative to not re-exposed control items) is similar (as in the current experiment). There is, therefore, precedent in the literature for situations in which RP yields no benefit over re-exposure. However, it is noteworthy that the retention intervals in RIF experiments tend to be relatively short (in the same experimental setting as the encoding and RP phase). Given evidence that RP effects emerge over the course of days [11], it may be that clearer RP effects would emerge in a RIF-type paradigm if the final test took place after several days. Notwithstanding this, our final test took place 2 days after the initial learning and retrieval/re-exposure phase, but we nevertheless observed no RP benefit over re-exposure.

Despite this lack of difference between the RP and re-exposure conditions, there was clear evidence for higher memory performance in both the RP and re-exposure conditions relative to the control condition. This suggests that both RP and re-exposure are able to improve memory retention and induce transfer. One possibility for a lack of difference between the RP and re-exposure conditions here is that retrieval was occurring during re-exposure. In other words, the re-exposure condition was sufficient to induce the benefits typically seen during RP alone. This explanation fits with the evidence for transfer in the re-exposure condition, suggesting that re-exposure led to the retrieval of all elements within a triplet. It is not clear whether this retrieval would have occurred automatically (e.g. via a more automatic pattern completion process during re-exposure) or due to an explicit strategy by the participants. However, participants were encouraged to 'encode' the re-exposed pairs, rather than retrieve associated information, and no participant reported using such an explicit strategy in the post-test questionnaire. Additionally, we would have expected such strategies to be more likely if participants had first experienced the RP condition before re-exposure, and our exploratory analyses revealed no effect of whether participants completed the RP or re-exposure condition first. It is, therefore, perhaps more likely that re-exposure for highly integrated triplets caused the relatively automatic retrieval of all triplet elements, resulting in increased retention for both re-exposed and not re-exposed material.

One possibility is that RP was similarly effective to re-exposure for the integrated triplets used in this study (compared to more typical RP material) as a consequence of a linear, as opposed to a nonlinear, forgetting function [67]. Critically, it has been argued that more complex, well integrated, event representations may follow a linear forgetting function relative to simple pairwise associations. Linear forgetting results in less forgetting, relative to a more typical nonlinear exponential decay function [68] early on in the forgetting process. Thus, linear forgetting is likely to present as increased retention relative to nonlinear forgetting, unless the retention interval is very long (e.g. several days/weeks). This means that the event triplets used here may be forgotten more slowly (in a linear fashion) relative to more typical pairwise associations used in previous RP experiments (e.g. [21,69]). This slower rate of forgetting may potentially decrease the extent to which benefits can be seen for RP relative to re-exposure, accounting for the lack of any difference between RP and re-exposure observed here. Experiments tracking forgetting for simple and complex stimuli (e.g. pairs versus triplets) following RP and re-exposure could be used to assess the influence of linear versus nonlinear forgetting on RP benefits.

Regardless of what is driving facilitation in both conditions, the benefits of RP *and* re-exposure, relative to the not re-exposed control condition, were clear. Previous RP studies sometimes, but not always, incorporate a not re-exposed control condition. When no such condition is included this may have the adverse effect of focusing attention on gains associated with RP relative to re-exposure, and as a consequence diminish the retention benefits of *both* RP and re-exposure. In short, although RP might be an optimal retention strategy in many situations, re-exposure can still facilitate long-term retention without the need for effortful retrieval. A greater focus on the effect sizes associated with re-

exposure versus no re-exposure and RP versus re-exposure in future studies would help to clarify the pedagogical value of these relative gains.

Methodological choices may have contributed to the reduced RP effect observed here. For example, we chose to use a cued recall test during RP, and multiple-choice during final test. Although multiple-choice final tests have been shown to produce medium-to-large effect sizes [6], and transfer from cued recall to multiple-choice also produces medium-to-large effect sizes [13], switching test format would likely result in a *reduced* effect size compared to if the two test formats had been identical. Importantly however, based on prior literature an RP versus re-exposure difference was still predicted.

Additionally, we rejected participants who achieved less than 20% cued recall accuracy which led to a high number of removed datasets ($N$ = 128). It is possible that there was, therefore, a qualitative difference between those participants who were rejected, and those who achieved greater than 20% recall accuracy and completed the final test. The literature on errorful generation suggests that not only is there a benefit of correctly recalling items but that there is also a benefit when items are incorrectly recalled [70,71]. This is not necessarily a reason to believe that the rejected participants would have benefited more from errorful generation than correct-answer generation, but future work could include these participants in the sample or at least reduce the threshold for determining low accuracy to investigate this possibility. On the other hand, the participants that remained in the sample still had overall relatively low accuracy during RP (43%) and, if the errorful generation effect played a role here, we would have expected to see an RP benefit in these participants.

Although it is possible that exclusion of low performers during RP may have reduced the RP relative to re-study effect, it is also the possible that the reverse is true—that the still relatively low performance of the included participants diminished the RP effect. For example, although induced by feedback at final test (which was not the case in the present study), there is evidence in the literature for a reverse-testing effect when performance during RP is low (e.g. [55]). Although participants did practice retrieval of each word-pair in an event twice in total, spaced and further repeated RP may have maximized the chances of finding an RP effect, in line with previous work that suggests the testing effect increases with multiple RP trials (e.g. [53–55]). However, and as noted in our earlier justification of the methodology, the RP effect has been found to be robust even with only a single trial [6].

Taken together, it seems unlikely that these methodological choices would have eliminated the RP effect entirely, but they may have played a role in reducing it. Future research is needed to systematically examine the influence of design elements on the RP effect, including investigating whether (and if so, when) re-exposure conditions can elicit automatic retrieval.

## 5.3. Cue versus target repetition

We also assessed whether the transfer effect was driven by the repetition of the cue or target within each triplet. No differences were seen between the *repeat cue–different target* trials in the RP relative to re-exposure condition, or the *different cue–repeat target* trials in the RP relative to re-exposure condition (analogous to the lack of difference between RP and re-exposure seen in the main analyses). We did see a difference between *repeat cue–different target* and *different cue–repeat target* trials for RP triplets, suggesting that repetition of the cue was more beneficial than repetition of the target. Although pre-registered, this analysis was theoretically agnostic in relation to whether cue or target repetition would be more beneficial to retention, and transfer appears to be present in both conditions (relative to the not re-exposed control condition). One unaddressed question is whether repetition of a cue or a target is necessary for transfer. To assess this, four elements would be needed (i.e. A–B–C–D), with RP for A–B pairs and final test for C–D pairs. If transfer is seen under these conditions, this would provide evidence for transfer in an integrated associative structure without the need for repetition of the cue or target between RP and final test.

## 6. Conclusion

To summarize, in the context of memory for event triplets of locations, objects and animals, we found evidence for improved retention following RP and re-exposure relative to no re-exposure. This improved retention was seen for both the tested/re-exposed pairs and the untested/not re-exposed pairs from the RP/re-exposed triplets. Thus, we provide evidence of transfer from repeated to not repeated pairs. Interestingly, we found no evidence for greater retention or transfer in the RP relative to the re-exposure condition, questioning the ubiquity of RP and highlighting the benefits of

re-exposure. It remains unclear whether this lack of difference was driven by an increased retention for re-exposure pairs, perhaps resulting from an automatic retrieval process during re-exposure or a lack of further facilitation from RP, perhaps driven by specific methodological choices. If the former, it suggests that re-exposure can be highly effective for retention under certain conditions (e.g. with relatively simple but highly integrated associative structures). If the latter, it suggests that effortful RP is not beneficial in all situations. In either case, the present findings suggest that presenting information in an integrated triplet format may have benefits for retention, encourage transfer, and may thus be pedagogically relevant. Further research is needed to uncover the underlying mechanisms driving these effects, to better inform the potential educational application of these findings.

Ethics. All participants provided informed consent prior to participating. The study was approved by the Department of Psychology's research ethics committee at the University of York (ref. 875).

Data accessibility. Fully anonymized data collected through Gorilla (i.e. the raw dataset with the qualitative answers for the exit questionnaire removed) as well as the data processing and analysis code are available on Github via the OSF (https://osf.io/bgm3p/) to allow researchers to reproduce our analysis, and the Gorilla Open Materials (https://app.gorilla.sc/openmaterials/107080) are available for researchers to replicate our study.

Authors' contributions. J.S.P. and A.J.H. conceived of the study, designed the study and drafted the manuscript. J.S.P. created the materials and statistical analyses, completed data collection and analysed the data. L.M.H. contributed to the design of the study and critically revised the manuscript. A.J.H. coordinated the study. All authors gave final approval for publication and agree to be held accountable for the work performed therein.

Competing interests. We declare we have no competing interests.

Funding. This research is funded by an Economic and Social Research Council (ESRC) grant awarded to A.J.H. and L.M.H. (grant no. ES/R007454/1). L.M.H. was additionally supported by ESRC grant no. ES/N009924/1.

Acknowledgements. We thank Emma James for her code that assisted with the processing of the data from Gorilla.

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
