## [Peer Review File · Royal Society Open Science]

Review History

RSOS-201456.R0 (Original submission)

Review form: Reviewer 1

Do you have any ethical concerns with this paper?

No

Recommendation?

Accept with minor revision

Comments to the Author(s)

Review of RSOS-201456: "Retrieval practice transfer effects for multielement event triplets"
Submitted to: Royal Society Open Science

Summary:

The degree to which retrieval practice enhances learning in cases wherein stimulus and response elements differ in some way on a final test has been the subject of recent investigation in the

literature (e.g., Pan et al., 2015). The authors propose to investigate whether the occurrence or nonoccurrence of enhanced learning in such cases is influenced by the degree with which the learned material is well-integrated during encoding. Participants will study multielement event triplets (e.g., animal-location-object), after which one-third of those triplets will be trained using retrieval practice (i.e., testing with feedback), another one-third will be trained using restudy, and the remaining one-third will not be trained at all. Approximately two days later, all participants will complete a final test that assesses retention (i.e., of previously tested stimulus-response associations) and transfer (i.e., of previously untested stimulus-response associations). The authors will then assess the degree to which an overlap (in the stimulus or response elements) between the practice test (i.e., retrieval practice) and final test influences performance on the final test.

1. The scientific validity of the research questions:

The proposed research promises to further advance scientific understanding of the effects of retrieval practice on memory, and specifically as it pertains to stimulus-response rearranged items. This research also has relevant connections to the retrieval-induced facilitation (e.g., Chan et al., 2006) and retrieval-induced forgetting (e.g., Anderson et al., 1994) literatures, as well as research on pattern completion (e.g., Horner et al., 2015). The research questions are scientifically valid.

2. The logic, rationale, and plausibility of the proposed hypotheses:

The proposed hypotheses, as enumerated in the text and in table form, are logical, well-reasoned, and plausible.

3. The soundness and feasibility of the methodology and analysis pipeline:

The methodology largely appears to be sound. The design closely follows that of the well-established retrieval practice literature, including an initial encoding phase, a retrieval practice / re-exposure phase, and a final test; the recruitment strategy uses Prolific, which is a fairly new but (thus far) high-quality online data-gathering platform; and the stimuli appear to be appropriate. The planned analyses appear to be appropriate as well.

One concern, however, involves the blocked nature of Phase 2. The current plan involves counterbalanced blocks of pure retrieval practice or pure restudy. There are concerns in the literature about possible strategy use that may occur when, for example, a retrieval practice block precedes a restudy block, or vice versa (although in some cases the use of such blocks appears to be immaterial, e.g., Carpenter et al., 2006, Experiment 2). Promisingly, the authors propose four blocks, such that not all restudied items will precede tested items, or vice versa. However, a more common alternative in the literature is to fully randomise the tested and restudied items, which the authors might consider (or at least provide a justification for avoiding). Alternatively, planned supplementary analyses that examine potential effects of block order could be conducted (which probably will reveal negligible effects).

In addition, the choice of a multiple-choice final test also bears some scrutiny. As Halamish and Bjork (2011) and others have demonstrated, an easier final test – that is, multiple-choice being easier than cued recall – might be less sensitive to testing effects (and cued recall tests are more impervious to effects of guessing). It is acceptable in this reviewer's view to use a multiple-choice final test, but the authors might want to consider test format when interpreting their results.

4. Whether the authors provide a sufficiently clear and detailed description of the methods to prevent undisclosed flexibility in the experimental procedures or analysis pipeline:

The methods are described sufficiently to prevent undisclosed flexibility (i.e., researcher degrees of freedom). Importantly, the data collection stopping and exclusion criteria are satisfactory.

5. Whether the authors have considered sufficient outcome-neutral conditions (e.g. positive controls) for ensuring that the results obtained are able to test the stated hypotheses:

The authors are using the well-established retrieval practice paradigm with re-exposure controls (i.e., restudy) as well as a no-training control.

Additional comments:

Please check some of the in-text citations that are missing, and a few that are oddly formatted. Further, the statement on p. 4, line 15, "...it has been proposed that practicing active retrieval creates a more elaborate memory trace and additional retrieval routes which in turn increase the likelihood of future retrieval," is accurate insofar as that is a prominent account of retrieval practice effects, but there are a variety of competing explanations; hence, that statement might be qualified somewhat.

Overall evaluation:

The proposed research is well thought-out and promises to reveal new insights into retrieval practice and transfer effects. I for one am excited to learn what the authors discover, and look forward to seeing this research carried out.

Review form: Reviewer 2

Do you have any ethical concerns with this paper?

No

Recommendation?

Accept with minor revision

Comments to the Author(s)

It is a well-developed, theoretically appropriately grounded experimental design with clear hypotheses. The feasibility of the methodology seems appropriate based on the plans. The methodological details and the planned analysis are clear and well-founded. The methodological details are adequately described so that the results of the research can be replicated. However, I would like to make a minor comment on the planned experimental design.

Since the main objective of the study is to examine whether after learning integrated information triples (e.g. location, famous person, object), retrieval practice (RP) of one piece of information (e.g. object) using the famous person as a cue would produce a memory transfer in the long run even for the non-recalled element (e.g. location). That is why, I think, I would like to draw the attention of the authors to an important aspect. The number of retrieval practice trials. Based on the research plan, I meant that the RP would consist of a single RP trial. Although the long-term positive effect of the test can be demonstrated for a single retrieval practice, there is a large difference between the single test and multiple tests situations in both the extent and nature of the effect and especially the degree of transfer. A recently published study examined a very similar issue using visual and verbal stimuli. Baddeley and colleagues applied two tests, one verbal, the Crimes Test, the other visual, the Four Doors Test. Each test involves four scenes comprising five features. They applied short-term and long-term delays and single test and

multiple tests conditions. Both the visual and verbal tests produced clear transfer for untested elements, however showed clear evidence of forgetting in the single test condition, together with little evidence of forgetting in the multi-test conditions.

See: Baddeley, A., Atkinson, A., Kemp, S., & Allen, R. (2019). The problem of detecting long-term forgetting: Evidence from the Crimes Test and the Four Doors Test. *Cortex*, 110, 69-79.

This result is in line with the latest results of our laboratory, repeated testing (more than three rounds and the rate of retrieval success significantly change the nature of factors that can be revealed in the background of the testing effect (retrieval speed, feedback effect, resistance to interference, etc.).

see:

Racsomány, M., Szöllősi, Á., & Bencze, D. (2018). Retrieval practice makes procedure from remembering: An automatization account of the testing effect. *Journal of Experimental Psychology: Learning, Memory, and Cognition*, 44(1), 157.

Pajkossy, P., Szöllősi, Á., & Racsomány, M. (2019). Retrieval practice decreases processing load of recall: Evidence revealed by pupillometry. *International Journal of Psychophysiology*, 143, 88-95.

Racsomány, M., Szöllősi, Á., & Marián, M. (2020). Reversing the testing effect by feedback is a matter of performance criterion at practice. *Memory & Cognition*. in press

This is why I would suggest the authors consider treating the number of RP rounds as an independent variable.

Decision letter (RSOS-201456.R0)

Dear Miss Pickering

On behalf of the Editors, I am pleased to inform you that your Manuscript RSOS-201456 entitled "Retrieval practice transfer effects for multielement event triplets" deemed suitable for in-principle acceptance in Royal Society Open Science subject to minor revision in accordance with the referee and editor suggestions. Please find their comments at the end of this email.

The reviewers and handling editors have recommended publication, but also suggest some minor revisions to your manuscript. Therefore, I invite you to respond to the comments and revise your manuscript.

Please you submit the revised version of your manuscript within 7 days (i.e. by the 24-Sep-2020). If you do not think you will be able to meet this date please let me know immediately.

To revise your manuscript, log into <https://mc.manuscriptcentral.com/rsos> and enter your Author Centre, where you will find your manuscript title listed under "Manuscripts with Decisions". Under "Actions," click on "Create a Revision." You will be unable to make your

revisions on the originally submitted version of the manuscript. Instead, revise your manuscript and upload a new version through your Author Centre.

Full author guidelines can be found here <https://royalsocietypublishing.org/rsos/registered-reports#ReviewerGuideRegRep>.

Kind regards
Mr Andrew Dunn
Royal Society Open Science
openscience@royalsociety.org

on behalf of Professor Chris Chambers (Subject Editor, Royal Society Open Science)
openscience@royalsociety.org

Associate Editor Comments to Author (Professor Chris Chambers):

Associate Editor: 1

Comments to the Author:

Two expert reviewers have now assessed the manuscript, and I'm happy to say that both provide positive recommendations. The reviewers are enthusiastic about the rationale and theoretical basis of the proposal, and the analysis plans, however they do offer some suggestions for revisions (or at least clarifications) to the study test procedures (including issues of blocking vs randomisation, recognition vs recall, and amount of retrieval practice). I invite the authors to consider these suggestions in a revised manuscript and response to the reviewers. Provided the authors respond thoroughly to all points raised, either revising the design or explaining the reasons for preferring the existing methodology, in-principle acceptance should be forthcoming without requiring further in-depth Stage 1 review.

Reviewer comments to Author:

Reviewer: 1

Comments to the Author(s)

Review of RSOS-201456: "Retrieval practice transfer effects for multielement event triplets"

Submitted to: Royal Society Open Science

Summary:

The degree to which retrieval practice enhances learning in cases wherein stimulus and response elements differ in some way on a final test has been the subject of recent investigation in the literature (e.g., Pan et al., 2015). The authors propose to investigate whether the occurrence or nonoccurrence of enhanced learning in such cases is influenced by the degree with which the learned material is well-integrated during encoding. Participants will study multielement event triplets (e.g., animal-location-object), after which one-third of those triplets will be trained using retrieval practice (i.e., testing with feedback), another one-third will be trained using restudy, and

the remaining one-third will not be trained at all. Approximately two days later, all participants will complete a final test that assesses retention (i.e., of previously tested stimulus-response associations) and transfer (i.e., of previously untested stimulus-response associations). The authors will then assess the degree to which an overlap (in the stimulus or response elements) between the practice test (i.e., retrieval practice) and final test influences performance on the final test.

1. The scientific validity of the research questions:

The proposed research promises to further advance scientific understanding of the effects of retrieval practice on memory, and specifically as it pertains to stimulus-response rearranged items. This research also has relevant connections to the retrieval-induced facilitation (e.g., Chan et al., 2006) and retrieval-induced forgetting (e.g., Anderson et al., 1994) literatures, as well as research on pattern completion (e.g., Horner et al., 2015). The research questions are scientifically valid.

2. The logic, rationale, and plausibility of the proposed hypotheses:

The proposed hypotheses, as enumerated in the text and in table form, are logical, well-reasoned, and plausible.

3. The soundness and feasibility of the methodology and analysis pipeline:

The methodology largely appears to be sound. The design closely follows that of the well-established retrieval practice literature, including an initial encoding phase, a retrieval practice / re-exposure phase, and a final test; the recruitment strategy uses Prolific, which is a fairly new but (thus far) high-quality online data-gathering platform; and the stimuli appear to be appropriate. The planned analyses appear to be appropriate as well.

One concern, however, involves the blocked nature of Phase 2. The current plan involves counterbalanced blocks of pure retrieval practice or pure restudy. There are concerns in the literature about possible strategy use that may occur when, for example, a retrieval practice block precedes a restudy block, or vice versa (although in some cases the use of such blocks appears to be immaterial, e.g., Carpenter et al., 2006, Experiment 2). Promisingly, the authors propose four blocks, such that not all restudied items will precede tested items, or vice versa. However, a more common alternative in the literature is to fully randomise the tested and restudied items, which the authors might consider (or at least provide a justification for avoiding). Alternatively, planned supplementary analyses that examine potential effects of block order could be conducted (which probably will reveal negligible effects).

In addition, the choice of a multiple-choice final test also bears some scrutiny. As Halamish and Bjork (2011) and others have demonstrated, an easier final test – that is, multiple-choice being easier than cued recall – might be less sensitive to testing effects (and cued recall tests are more impervious to effects of guessing). It is acceptable in this reviewer's view to use a multiple-choice final test, but the authors might want to consider test format when interpreting their results.

4. Whether the authors provide a sufficiently clear and detailed description of the methods to prevent undisclosed flexibility in the experimental procedures or analysis pipeline:

The methods are described sufficiently to prevent undisclosed flexibility (i.e., researcher degrees of freedom). Importantly, the data collection stopping and exclusion criteria are satisfactory.

5. Whether the authors have considered sufficient outcome-neutral conditions (e.g. positive controls) for ensuring that the results obtained are able to test the stated hypotheses:

The authors are using the well-established retrieval practice paradigm with re-exposure controls (i.e., restudy) as well as a no-training control.

Additional comments:

Please check some of the in-text citations that are missing, and a few that are oddly formatted. Further, the statement on p. 4, line 15, "...it has been proposed that practicing active retrieval creates a more elaborate memory trace and additional retrieval routes which in turn increase the likelihood of future retrieval," is accurate insofar as that is a prominent account of retrieval practice effects, but there are a variety of competing explanations; hence, that statement might be qualified somewhat.

Overall evaluation:

The proposed research is well thought-out and promises to reveal new insights into retrieval practice and transfer effects. I for one am excited to learn what the authors discover, and look forward to seeing this research carried out.

Reviewer: 2

Comments to the Author(s)

It is a well-developed, theoretically appropriately grounded experimental design with clear hypotheses. The feasibility of the methodology seems appropriate based on the plans. The methodological details and the planned analysis are clear and well-founded. The methodological details are adequately described so that the results of the research can be replicated. However, I would like to make a minor comment on the planned experimental design.

Since the main objective of the study is to examine whether after learning integrated information triples (e.g. location, famous person, object), retrieval practice (RP) of one piece of information (e.g. object) using the famous person as a cue would produce a memory transfer in the long run even for the non-recalled element (e.g. location). That is why, I think, I would like to draw the attention of the authors to an important aspect. The number of retrieval practice trials. Based on the research plan, I meant that the RP would consist of a single RP trial. Although the long-term positive effect of the test can be demonstrated for a single retrieval practice, there is a large difference between the single test and multiple tests situations in both the extent and nature of the effect and especially the degree of transfer. A recently published study examined a very similar issue using visual and verbal stimuli. Baddeley and colleagues applied two tests, one verbal, the Crimes Test, the other visual, the Four Doors Test. Each test involves four scenes comprising five features. They applied short-term and long-term delays and single test and multiple tests conditions. Both the visual and verbal tests produced clear transfer for untested elements, however showed clear evidence of forgetting in the single test condition, together with little evidence of forgetting in the multi-test conditions.

See: Baddeley, A., Atkinson, A., Kemp, S., & Allen, R. (2019). The problem of detecting long-term forgetting: Evidence from the Crimes Test and the Four Doors Test. *Cortex*, 110, 69-79.

This result is in line with the latest results of our laboratory, repeated testing (more than three rounds and the rate of retrieval success significantly change the nature of factors that can be revealed in the background of the testing effect (retrieval speed, feedback effect, resistance to interference, etc.).

see:

Racsmány, M., Szóllósi, Á., & Bencze, D. (2018). Retrieval practice makes procedure from remembering: An automatization account of the testing effect. *Journal of Experimental Psychology: Learning, Memory, and Cognition*, 44(1), 157.

Pajkossy, P., Szóllósi, Á., & Racsmány, M. (2019). Retrieval practice decreases processing load of recall: Evidence revealed by pupillometry. *International Journal of Psychophysiology*, 143, 88-95.

Racsmány, M., Szóllósi, Á., & Marián, M. (2020). Reversing the testing effect by feedback is a matter of performance criterion at practice. *Memory & Cognition*. in press

This is why I would suggest the authors consider treating the number of RP rounds as an independent variable.

Author's Response to Decision Letter for (RSOS-201456.R0)

See Appendix A.

Decision letter (RSOS-201456.R1)

Dear Jade

On behalf of the Editor, I am pleased to inform you that your Manuscript RSOS-201456.R1 entitled "Retrieval practice transfer effects for multielement event triplets" has been accepted in principle for publication in Royal Society Open Science.

You may now progress to Stage 2 and complete the study as approved. Before commencing data collection we ask that you:

- 1) Update the journal office as to the anticipated completion date of your study.
- 2) Register your approved protocol on the Open Science Framework (e.g. using the dedicated RR registration portal at <https://osf.io/rr>) or other recognised repository, either publicly or privately under embargo until submission of the Stage 2 manuscript. Please note that a time-stamped, independent registration of the protocol is mandatory under journal policy, and manuscripts that do not conform to this requirement cannot be considered at Stage 2. The protocol should be registered unchanged from its current approved state, with the time-stamp preceding implementation of the approved study design.

Following completion of your study, we invite you to resubmit your paper for peer review as a Stage 2 Registered Report. Please note that your manuscript can still be rejected for publication at Stage 2 if the Editors consider any of the following conditions to be met:

- The results were unable to test the authors' proposed hypotheses by failing to meet the approved outcome-neutral criteria.
- The authors altered the Introduction, rationale, or hypotheses, as approved in the Stage 1 submission.
- The authors failed to adhere closely to the registered experimental procedures. Please note that any deviations from the approved experimental procedures must be communicated to the editor immediately for approval, and prior to the completion of data collection. Failure to do so can result in revocation of in-principle acceptance and rejection at Stage 2 (see complete guidelines for further information).
- Any post-hoc (unregistered) analyses were either unjustified, insufficiently caveated, or overly dominant in shaping the authors' conclusions.
- The authors' conclusions were not justified given the data obtained.

We encourage you to read the complete guidelines for authors concerning Stage 2 submissions at <https://royalsocietypublishing.org/rsos/registered-reports#ReviewerGuideRegRep>. Please especially note the requirements for data sharing, reporting the URL of the independently registered protocol, and that withdrawing your manuscript will result in publication of a Withdrawn Registration.

Please note that Royal Society Open Science will introduce article processing charges for all new submissions received from 1 January 2018. Registered Reports submitted and accepted after this date will ONLY be subject to a charge if they subsequently progress to and are accepted as Stage 2 Registered Reports. If your manuscript is submitted and accepted for publication after 1 January 2018 (i.e. as a full Stage 2 Registered Report), you will be asked to pay the article processing charge, unless you request a waiver and this is approved by Royal Society Publishing. You can find out more about the charges at <https://royalsocietypublishing.org/rsos/charges>. Should you have any queries, please contact openscience@royalsociety.org.

Once again, thank you for submitting your manuscript to Royal Society Open Science and we look forward to receiving your Stage 2 submission. If you have any questions at all, please do not hesitate to get in touch. We look forward to hearing from you shortly with the anticipated submission date for your stage two manuscript.

on behalf of Professor Chris Chambers (Registered Reports Editor, Royal Society Open Science)
openscience@royalsociety.org

Author's Response to Decision Letter for (RSOS-201456.R1)

See Appendix B.

RSOS-201456.R2

Review form: Reviewer 1

Is the manuscript scientifically sound in its present form?

Yes

Are the interpretations and conclusions justified by the results?

Yes

Is the language acceptable?

Yes

Do you have any ethical concerns with this paper?

No

Have you any concerns about statistical analyses in this paper?

No

Recommendation?

Accept with minor revision

Comments to the Author(s)

Review of RSOS-201456.R2: "Retrieval practice transfer effects for multielement event triplets"
Submitted to: Royal Society Open Science

Overall comment:

I read the Stage 2 manuscript with interest and was happy to see that the authors successfully executed a carefully-controlled study that aligned with their Stage 1 plans. Overall, this manuscript makes a useful contribution to the retrieval practice and transfer literatures, as well as related literatures, and I commend the authors on their work. It was also very encouraging to see the publicly available materials and ability to preview the study itself – the availability of such resources will be a great help to anyone wishing to replicate and/or extend this work.

Two interpretative issues that the authors may consider:

First, from an educational standpoint, the utility of "transfer" relative to a non-exposed/no-reexposure condition is debatable. The more meaningful comparison arguably is that versus a re-exposure condition (or some form of "genuine" control activity), which the authors correctly include. On that front, the fact that no transfer was observed relative to a re-exposure condition is highly consistent with the prior literature addressing retrieval practice and stimulus-response transfer.

Second, the absence of a retrieval practice effect invites additional scrutiny of the study design (especially given the almost ubiquitous nature of retrieval practice effects). This reviewer had raised concerns about the use of a multiple-choice final test given prior evidence that such tests are less sensitive at detecting retrieval practice effects than, for example, a short answer test. It is understood that a more easily scored test format was ultimately chosen for logistical reasons, but the final test format might be considered as a factor in the finding of a statistically insignificant retrieval practice effect.

As an additional point, one raises an eyebrow at the removal of 128 participants due to low accuracy during cued recall. Ultimately, this is not necessarily a major problem. However, it raises the possibility that the participants that were retained in the study were qualitatively different in some way.

Further comments:

1. Whether the data are able to test the authors' proposed hypotheses by passing the approved outcome-neutral criteria.

Yes, the data are able to test the proposed hypotheses.

2. Whether the Introduction, rationale and stated hypotheses are the same as the approved Stage 1 submission.

Yes, the Stage 1 and Stage 2 manuscripts are aligned on those aspects.

3. Whether the authors adhered precisely to the registered experimental procedures.

Yes, the authors adhered to the experimental procedures with negligible deviation (a small error occurred on the final test, but as discussed on p. 20, did not appreciably impact the results or conclusions).

4. Where applicable, whether any unregistered exploratory statistical analyses are justified, methodologically sound, and informative.

The exploratory analyses are clearly identified, methodologically appropriate, reasonable, and informative.

5. Whether the authors' conclusions are justified given the data.

Yes, the conclusions are largely justified.

Review form: Reviewer 2

Is the manuscript scientifically sound in its present form?

No

Are the interpretations and conclusions justified by the results?

Yes

Is the language acceptable?

Yes

Do you have any ethical concerns with this paper?

No

Have you any concerns about statistical analyses in this paper?

No

Recommendation?

Major revision

Comments to the Author(s)

The authors have implemented their research plan very thoroughly, providing detailed and correct analyses of the results. The main results are: in the final test after a two-day delay, the previous retrieval practice showed a memory advantage over the baseline control condition (no retrieval, no re-exposure) in the multiple-choice test, but no advantage over the re-exposure condition. In terms of transfer effect, both retrieval practice and re-exposure showed an advantage over the baseline control condition, whereas retrieval practice showed no difference compared to the re-exposure condition.

As this is a pre-registered research and I have already evaluated the research plans in the first round, so without repeating what was described there in detail, I can easily articulate my main problem with the study.

Earlier I indicated that a single round of presentation on 54 triplets and then a single round of retrieval practice I did not find reassuring in terms of examining the testing effect and especially the transfer effect of retrieval practice. My main problem even then was that although there are results in the testing effect literature that have a testing effect even with low retrieval success, it has been repeatedly shown that the effect is less robust and sometimes reversed testing effect (better memory performance in the re-exposure condition) was found. The authors nevertheless decided not to change the design.

The results confirmed my preliminary fears. Recall performance during practice is quite low, if I interpret the data in Table 1 well, the recall success rate of the event triplets was 43% in the practice phase, which was accompanied by a fairly high false event recall (37%). Together, we can say that the subjects were able to recall less than half of the event information correctly and provided wrong event information in every third recall trials. Since we are talking here about an episodic memory test built on triplet learning, low retrieval success combined with a significant number of false recalls cannot be considered an optimal situation to analyse the long-term effects of retrieval practice.

This is also reflected in the final results. As a testing effect, the literature primarily refers to the long-term advantage of retrieval practice over simple re-exposure in the literature, not the long-term advantage over no-exposure. It has been demonstrated countless times that retrieval practice produces a robust long-term memory advantage over re-exposure, so its lack is presumably due to the design problems discussed earlier. Since retrieval practice does not show an advantage over re-exposure in the final test, it is questionable that any difference between the two conditions would be expected in the transfer test. This problem, I think, significantly reduces the clarity of the results. Notably, it is difficult to say that the results reveal a boundary condition related to transfer, or only draw attention to the fact that low retrieval success during retrieval practice, combined with a significant amount of false recall, eliminates the testing effect in both direct recall and in the transfer effect.

I see two ways to remedy this problem. 1) running a new experiment on a modified design, taking into account the consequences of this study. 2) writing a detailed limitation paragraph in the discussion where the problems I have described are explored.

Decision letter (RSOS-201456.R2)

Dear Jade:

On behalf of the Editor, I am pleased to inform you that your Stage 2 Registered Report RSOS-201456.R2 entitled "Retrieval practice transfer effects for multielement event triplets" has been deemed suitable for publication in Royal Society Open Science subject to minor revision in accordance with the referee suggestions. Please find the referees' comments at the end of this email.

The reviewers and Subject Editor have recommended publication, but also suggest some minor revisions to your manuscript. Therefore, I invite you to respond to the comments and revise your manuscript.

Please also ensure that all the below editorial sections are included where appropriate -- if any section is not applicable to your manuscript, please can we ask you to nevertheless include the heading, but explicitly state that the heading is inapplicable. An example of these sections is attached with this email.

- Ethics statement

- Data accessibility

[http://datadryad.org/submit?journalID=RSOS&manu=\(Document not available\)](http://datadryad.org/submit?journalID=RSOS&manu=(Document not available))

- Competing interests

- Authors' contributions

All submissions, other than those with a single author, must include an Authors' Contributions section which individually lists the specific contribution of each author. The list of Authors should meet all of the following criteria; 1) substantial contributions to conception and design, or

acquisition of data, or analysis and interpretation of data; 2) drafting the article or revising it critically for important intellectual content; and 3) final approval of the version to be published.

- Acknowledgements

- Funding statement

Because the schedule for publication is very tight, it is a condition of publication that you submit the revised version of your manuscript within 7 days (i.e. by the 30-Sep-2021). If you do not think you will be able to meet this date please let me know immediately.

on behalf of Professor Chris Chambers
(Registered Reports Editor, Royal Society Open Science)
openscience@royalsociety.org

Associate Editor Comments to Author (Professor Chris Chambers):

Comments to the Author:

The Stage 2 manuscript was kindly evaluated by the two original Stage 1 reviewers. The reviewers agree that the manuscript broadly meets the Stage 2 criteria, while also offering some critical points and suggestions concerning the methodology. At Stage 2 the preregistered methodology of a Registered Report is not formally reassessed (regardless of whether or not the reviewers were perfectly satisfied with the methods at the conclusion of the Stage 1 review process); therefore these comments cannot lead to an editorial rejection or requirement for further experiments. However, they do prompt the need for careful consideration in either (or both) the response to reviewers (via rebuttal) and the Stage 2 Discussion. Provided the authors are able to respond comprehensively to all points raised, final acceptance should be forthcoming without requiring further in-depth review.

Comments to Author:

Reviewer: 1

Comments to the Author(s)

Review of RSOS-201456.R2: "Retrieval practice transfer effects for multielement event triplets"

Submitted to: Royal Society Open Science

Overall comment:

I read the Stage 2 manuscript with interest and was happy to see that the authors successfully executed a carefully-controlled study that aligned with their Stage 1 plans. Overall, this manuscript makes a useful contribution to the retrieval practice and transfer literatures, as well as related literatures, and I commend the authors on their work. It was also very encouraging to see the publicly available materials and ability to preview the study itself – the availability of such resources will be a great help to anyone wishing to replicate and/or extend this work.

Two interpretative issues that the authors may consider:

First, from an educational standpoint, the utility of "transfer" relative to a non-exposed/no-reexposure condition is debatable. The more meaningful comparison arguably is that versus a re-

exposure condition (or some form of “genuine” control activity), which the authors correctly include. On that front, the fact that no transfer was observed relative to a re-exposure condition is highly consistent with the prior literature addressing retrieval practice and stimulus-response transfer.

Second, the absence of a retrieval practice effect invites additional scrutiny of the study design (especially given the almost ubiquitous nature of retrieval practice effects). This reviewer had raised concerns about the use of a multiple-choice final test given prior evidence that such tests are less sensitive at detecting retrieval practice effects than, for example, a short answer test. It is understood that a more easily scored test format was ultimately chosen for logistical reasons, but the final test format might be considered as a factor in the finding of a statistically insignificant retrieval practice effect.

As an additional point, one raises an eyebrow at the removal of 128 participants due to low accuracy during cued recall. Ultimately, this is not necessarily a major problem. However, it raises the possibility that the participants that were retained in the study were qualitatively different in some way.

Further comments:

1. Whether the data are able to test the authors’ proposed hypotheses by passing the approved outcome-neutral criteria.

Yes, the data are able to test the proposed hypotheses.

2. Whether the Introduction, rationale and stated hypotheses are the same as the approved Stage 1 submission.

Yes, the Stage 1 and Stage 2 manuscripts are aligned on those aspects.

3. Whether the authors adhered precisely to the registered experimental procedures.

Yes, the authors adhered to the experimental procedures with negligible deviation (a small error occurred on the final test, but as discussed on p. 20, did not appreciably impact the results or conclusions).

4. Where applicable, whether any unregistered exploratory statistical analyses are justified, methodologically sound, and informative.

The exploratory analyses are clearly identified, methodologically appropriate, reasonable, and informative.

5. Whether the authors’ conclusions are justified given the data.

Yes, the conclusions are largely justified.

Reviewer: 2

Comments to the Author(s)

The authors have implemented their research plan very thoroughly, providing detailed and correct analyses of the results. The main results are: in the final test after a two-day delay, the previous retrieval practice showed a memory advantage over the baseline control condition (no retrieval, no re-exposure) in the multiple-choice test, but no advantage over the re-exposure

condition. In terms of transfer effect, both retrieval practice and re-exposure showed an advantage over the baseline control condition, whereas retrieval practice showed no difference compared to the re-exposure condition.

As this is a pre-registered research and I have already evaluated the research plans in the first round, so without repeating what was described there in detail, I can easily articulate my main problem with the study.

Earlier I indicated that a single round of presentation on 54 triplets and then a single round of retrieval practice I did not find reassuring in terms of examining the testing effect and especially the transfer effect of retrieval practice. My main problem even then was that although there are results in the testing effect literature that have a testing effect even with low retrieval success, it has been repeatedly shown that the effect is less robust and sometimes reversed testing effect (better memory performance in the re-exposure condition) was found. The authors nevertheless decided not to change the design.

The results confirmed my preliminary fears. Recall performance during practice is quite low, if I interpret the data in Table 1 well, the recall success rate of the event triplets was 43% in the practice phase, which was accompanied by a fairly high false event recall (37%). Together, we can say that the subjects were able to recall less than half of the event information correctly and provided wrong event information in every third recall trials. Since we are talking here about an episodic memory test built on triplet learning, low retrieval success combined with a significant number of false recalls cannot be considered an optimal situation to analyse the long-term effects of retrieval practice.

This is also reflected in the final results. As a testing effect, the literature primarily refers to the long-term advantage of retrieval practice over simple re-exposure in the literature, not the long-term advantage over no-exposure. It has been demonstrated countless times that retrieval practice produces a robust long-term memory advantage over re-exposure, so its lack is presumably due to the design problems discussed earlier. Since retrieval practice does not show an advantage over re-exposure in the final test, it is questionable that any difference between the two conditions would be expected in the transfer test. This problem, I think, significantly reduces the clarity of the results. Notably, it is difficult to say that the results reveal a boundary condition related to transfer, or only draw attention to the fact that low retrieval success during retrieval practice, combined with a significant amount of false recall, eliminates the testing effect in both direct recall and in the transfer effect.

I see two ways to remedy this problem. 1) running a new experiment on a modified design, taking into account the consequences of this study. 2) writing a detailed limitation paragraph in the discussion where the problems I have described are explored.

Author's Response to Decision Letter for (RSOS-201456.R2)

See Appendix C.

Decision letter (RSOS-201456.R3)

Dear Jade

It is a pleasure to accept your manuscript entitled "Retrieval practice transfer effects for multielement event triplets" in its current form for publication in Royal Society Open Science. The comments of the reviewer(s) who reviewed your manuscript are included at the foot of this letter.

Please ensure that you send to the editorial office individual files for each figure and table included in your manuscript. You can send these in a zip folder if more convenient. Failure to provide these files may delay the processing of your proof. You may disregard this request if you have already provided these files to the editorial office.

on behalf of Professor Chris Chambers (Subject Editor)
openscience@royalsociety.org

Appendix A

We thank the reviewers and editor for their helpful comments and input. We have addressed all their comments, and believe the manuscript is ready for Stage 1 acceptance. Responses to comments are in **bold**, and additions to the manuscript are in *italics*.

Reviewer 1

1. One concern, however, involves the blocked nature of Phase 2. The current plan involves counterbalanced blocks of pure retrieval practice or pure restudy. There are concerns in the literature about possible strategy use that may occur when, for example, a retrieval practice block precedes a restudy block, or vice versa (although in some cases the use of such blocks appears to be immaterial, e.g., Carpenter et al., 2006, Experiment 2). Promisingly, the authors propose four blocks, such that not all restudied items will precede tested items, or vice versa. However, a more common alternative in the literature is to fully randomise the tested and restudied items, which the authors might consider (or at least provide a justification for avoiding). Alternatively, planned supplementary analyses that examine potential effects of block order could be conducted (which probably will reveal negligible effects).

We did consider a fully randomised design, but we are concerned that introducing an element of task-switching may cloud the processes of retrieval vs re-exposure which we want to keep as distinct as possible. Secondly, this may introduce unnecessary added difficulty for participants which we want to keep to a minimum given the online nature of the study. In our experience, participants are much less likely to ask questions to clarify task instructions through Prolific than they are when the researcher is in the room with them. As noted, we did try to reduce order effects with the proposed ABAB design (or BABA for the other half of participants) instead of a simple AB (or BA) design. As the study by Carpenter et al. (2006) notes, it is unlikely to affect our results. We have added an explanatory section in our method to reflect our considerations on this point.

p.11: “We opted for a blocked design rather than randomising the conditions trial by trial to reduce effects of task-switching, and to reduce any difficulty for participants in comprehending task instructions in the online environment where they are less likely to ask questions of the researchers for clarity. Previous lab-based research suggests RP is robust in both a mixed and blocked design (e.g. see Carpenter et al., 2006).”

A future direction for this line of research could examine strategy use in relation to blocked designs.

2. In addition, the choice of a multiple-choice final test also bears some scrutiny. As Halamish and Bjork (2011) and others have demonstrated, an easier final test – that is, multiple-choice being easier than cued recall – might be less sensitive to testing effects (and cued recall tests are more impervious to effects of guessing). It is acceptable in this reviewer’s view to use a multiple-choice final test, but the authors might want to consider test format when interpreting their results.

We take on board the concerns about opting for multiple choice over cued recall for the final test format. Our rationale was primarily based on logistics. Whilst we have tried to partially automate the process of checking the accuracy of cued recall responses during retrieval practice blocks using

R, many answers will still require two independent researchers to manually check the accuracy of the remaining trials. In Phase 3 (the final test) this amounts to 324 trials per participant, and our planned sample size ($n = 112$) means that the number of trials requiring manual screening will be substantial in number and difficult to manage. As the reviewer suggests, we will keep this choice of testing format in mind when interpreting the results. Whilst multiple choice may be relatively less sensitive to testing effects, the literature consistently demonstrates testing effects using multiple-choice tests, which we have reflected in a small addition to the manuscript:

p.11: *“Multiple-choice final tests have been shown to produce medium-to-large effect sizes in a recent meta-analysis (Adesope et al., 2017).”*

3. Please check some of the in-text citations that are missing, and a few that are oddly formatted. Further, the statement on p. 4, line 15, "...it has been proposed that practicing active retrieval creates a more elaborate memory trace and additional retrieval routes which in turn increase the likelihood of future retrieval," is accurate insofar as that is a prominent account of retrieval practice effects, but there are a variety of competing explanations; hence, that statement might be qualified somewhat.

We have fixed the formatting issues with existing citations (i.e. incorrect meta-data in Zotero had meant that the initials and first names of authors were incorrectly included in the in-text citations). The reviewer also noted that some in-text citations are missing. We were unable to identify these instances and so have not added any extra citations.

Finally, we have also qualified the statement on p4, line 15, to make it clear that this is one competing explanation, and have added further citations related to competing theories:

“The underlying mechanisms of RP effects are less clear, but one proposal suggests that practicing active retrieval creates a more elaborate memory trace and additional retrieval routes which in turn increase the likelihood of future retrieval (Roediger & Butler, 2011, but for alternative accounts see also e.g. Adesope et al., 2017; Antony et al., 2017).”

Reviewer 2:

1. Since the main objective of the study is to examine whether after learning integrated information triples (e.g. location, famous person, object), retrieval practice (RP) of one piece of information (e.g. object) using the famous person as a cue would produce a memory transfer in the long run even for the non-recalled element (e.g. location). That is why, I think, I would like to draw the attention of the authors to an important aspect. The number of retrieval practice trials. Based on the research plan, I meant that the RP would consist of a single RP trial. Although the long-term positive effect of the test can be demonstrated for a single retrieval practice, there is a large difference between the single test and multiple tests situations in both the extent and nature of the effect and especially the degree of transfer. A recently published study examined a very similar issue using visual and verbal stimuli. Baddeley and colleagues applied two tests, one verbal, the Crimes Test, the other visual, the Four Doors Test. Each test involves four scenes comprising five features. They applied short-term and long-term delays and single test and multiple tests conditions. Both the visual and verbal tests produced clear

transfer for untested elements, however showed clear evidence of forgetting in the single test condition, together with little evidence of forgetting in the multi-test conditions. This result is in line with the latest results of our laboratory, repeated testing (more than three rounds and the rate of retrieval success significantly change the nature of factors that can be revealed in the background of the testing effect (retrieval speed, feedback effect, resistance to interference, etc.). This is why I would suggest the authors consider treating the number of RP rounds as an independent variable.

Thanks for drawing our attention to the Baddeley et al (2019) paper, we've incorporated this citation into our manuscript.

We note that the main concern is the use of a single retrieval practice trial per triplet rather than multiple trials, and that this is based on several recent research studies which suggest that multiple (three or more) retrieval practice trials result in a stronger testing effect.

First, it is important to note that participants will perform two retrieval practice trials per triplet as each association is tested in both directions. For example, if *spider – circus* is tested in the first block of retrieval practice trials, *circus – spider* is tested in the second block of retrieval practice trials. We have realised this information is not immediately accessible in the text and so we have made minor edits to the manuscript for clarity.

e.g., p. 6: *“Importantly, for both the RP and re-exposure condition, only one pairwise association per triplet will be tested/re-exposed (although each association will be tested twice in total; once in both directions over two separate blocks).”*

A recent meta-analysis of the testing effect suggested that one practice trial is sufficient to produce a testing effect with a large effect size (0.7; Adesope et al., 2017). For the current study we are working with limited financial resources and doubling the number of Phase 2 trials (in order to have two trials per association direction, and thus four trials total) would increase the costs of participant payment to more than our budget allows if we want to reach the planned sample size according to our power calculations. We think that this is an important point to keep in mind though, and that the effects of multiple (three or more) trials on transfer effects is a fruitful avenue for future investigation. We have added some discussion of this into the manuscript:

p. 10: *“Therefore, each tested/re-exposed association is seen twice in total during this phase. Although there is evidence to suggest that the testing effect may increase with multiple RP trials (e.g. Baddeley et al., 2019; Pajkossy et al., 2019; Racsomány et al., 2020), the effect has still been established to be robust with only a single trial (Adesope et al., 2017).”*

Additional changes

We piloted the procedure and analysis on a small number of participants (in the interests of transparency this data will not form part of the final dataset and was solely to test the procedure) which highlighted some areas in the manuscript that lacked clarity, and so we have made some minor improvements:

- 1. Fluent, but non-native, English speakers reported difficulty with understanding and remembering the word stimuli, so we have changed our inclusion criteria from fluent English speakers to native English speakers.**
- 2. We have added some clarifications to the “Data processing” procedure, including how we plan to categorise errors, how much of this process will be automated vs manually rated by the researchers.**
- 3. We have clarified in the “Data exclusion” section at which point we will rate the overall accuracy for exclusion. We will exclude very low performers before we check for typographical errors and manually rate responses, rather than after.**

Appendix B

Dear Professor Chris Chambers (subject editor for Registered Reports, Royal Society Open Science),

Please find included our Stage 2 Registered Report “Retrieval practice transfer effects for multielement event triplets”. Despite a robust literature on the topic, we did not replicate a retrieval practice effect over and above re-exposure, which calls for further investigation into the boundary conditions of the effect. We found, however, that both retrieval practice and re-exposure induce transfer from tested/re-exposed information to untested/not re-exposed information, at least when the material is highly integrated.

The Introduction and Methods sections are unchanged from the Stage 1 accepted Registered Report except to transform it from the future tense to the past tense, add links to relevant repositories, and to fix typos that went previously unseen. We also changed Figure 1 aesthetically to match our plots, although the content itself remains the same. We have uploaded an additional version of the manuscript that contains tracked changes for the content from the Stage 1 manuscript for full transparency.

The project is funded by the Economic and Social Research Council (ES/R007454/1) and ethical approval has been received from the Department of Psychology Ethics Committee at the University of York (ref: 875). On receipt of “in principle acceptance”, the approved Stage 1 registered report protocol was uploaded to the OSF (<https://osf.io/qgah7>; link on p13 of the manuscript) before data collection began. All data in this manuscript was collected after IPA was received and the protocol uploaded to the OSF. Stimuli and power analyses are on the OSF (<https://osf.io/wtyku/>; link on p8 of the manuscript), and all (de-identified) raw data and code for analyses are now available on Github (<https://github.com/jspickering/registered-report-retrieval-transfer>, linked via the OSF page <https://osf.io/bgm3p/> on p14 of the manuscript), and the full experiment is available via Gorilla Open Materials (<https://app.gorilla.sc/openmaterials/107080>; link on p9 and 14 of the manuscript).

Best wishes,

Jade S Pickering,

Lisa-Marie Henderson,

Aidan J Horner

Appendix C

We thank the reviewers for their kind words and helpful comments on our Stage 2 manuscript. We have addressed all comments where appropriate, and believe the manuscript is ready for Stage 2 acceptance. Responses to comments are in **bold**, and additions to the manuscript are in *italics*. Changes to the manuscript are highlighted in yellow on a separate (pdf) version of the document.

Both reviewers highlight methodological concerns in context of the lack of the expected RP effect, which were originally raised during Stage 1 reviews. The Stage 1 comments were in relation to **maximising** the RP effect, and no a priori doubt was present from either reviewer or the study authors about whether an RP effect should have been **present at all**. We cited literature that suggested our design should still produce an RP effect and included several manipulations that tried to maximise the effect. We therefore believe, based on prior literature, an effect should have been seen. We address these concerns in more detail below.

Reviewer 1

1. First, from an educational standpoint, the utility of “transfer” relative to a non-exposed/no-reexposure condition is debatable. The more meaningful comparison arguably is that versus a re-exposure condition (or some form of “genuine” control activity), which the authors correctly include. On that front, the fact that no transfer was observed relative to a re-exposure condition is highly consistent with the prior literature addressing retrieval practice and stimulus-response transfer.

We agree with the reviewer. We believe the previous submitted version of the manuscript was clear in relation to transfer relative to the not re-exposed baseline condition, and the conclusions were appropriate based on this finding. The question of “utility” in an educational environment is more difficult to quantify or define, however our view is that transfer is important otherwise the RP effect would be highly specific to the exact conditions under which RP was carried out. This would decrease the real-world relevance of RP.

2. Second, the absence of a retrieval practice effect invites additional scrutiny of the study design (especially given the almost ubiquitous nature of retrieval practice effects). This reviewer had raised concerns about the use of a multiple-choice final test given prior evidence that such tests are less sensitive at detecting retrieval practice effects than, for example, a short answer test. It is understood that a more easily scored test format was ultimately chosen for logistical reasons, but the final test format might be considered as a factor in the finding of a statistically insignificant retrieval practice effect.

We expected that a multiple choice test would result in a slightly reduced testing effect compared to a cued recall final test but we do not believe there were any a priori reasons to suggest it would lead to a statistically insignificant retrieval practice effect. Indeed, the reviewer originally stated this manipulation was appropriate: “It is acceptable in this reviewer’s view to use a multiple-choice final test, but the authors might want to consider test format when interpreting their results.” We agree that this is an important point when interpreting our results though (see below for additional text).

A recent comprehensive meta-analysis found that transfer from cued recall to multiple-choice shows a medium-to-large effect size (Pan & Rickard, 2018) and a meta-analysis of retrieval practice

effects generally finds medium-to-large effect sizes for multiple-choice final tests (Adesope et al., 2017). Our interpretation of these results was not that we didn't find a retrieval practice effect, but that the re-exposure control was not suitable and may have elicited automatic retrieval. However, we think it is important to consider all possibilities and so we have added more discussion of test-format in an additional paragraph that discusses methodological choices more generally.

p22: *“Methodological choices may have contributed to the reduced RP effect observed here. For example, we chose to use a cued recall test during retrieval practice, and multiple-choice during final test. Although multiple-choice final tests have been shown to produce medium-to-large effect sizes (Adesope et al., 2017), and transfer from cued recall to multiple-choice also produces medium-to-large effect sizes (Pan & Rickard, 2018), switching test format would likely result in a reduced effect size compared to if the two test formats had been identical. Importantly however, based on prior literature an RP versus re-exposure difference was still predicted.”*

3. As an additional point, one raises an eyebrow at the removal of 128 participants due to low accuracy during cued recall. Ultimately, this is not necessarily a major problem. However, it raises the possibility that the participants that were retained in the study were qualitatively different in some way.

We were surprised at the number of participants that needed to be removed for low accuracy during cued recall and agree that there may have been some qualitative differences between these participants and those that went forward to the final test. We have added some discussion of this, and referenced the literature on error generation.

p23: *“Additionally, we rejected participants who achieved <20% cued recall accuracy which led to a high number of removed datasets (N = 128). It is possible that there was therefore a qualitative difference between those participants who were rejected, and those who achieved >20% recall accuracy and completed the final test. The literature on errorful generation suggests that not only is there a benefit of correctly recalling items but that there is also a benefit when items are incorrectly recalled (Potts, Davies, & Shanks, 2019; Potts & Shanks, 2014). This is not necessarily a reason to believe that the rejected participants would have benefited more from errorful generation than correct-answer generation, but future work could include these participants in the sample or at least reduce the threshold for determining low accuracy to investigate this possibility. On the other hand, the participants that remained in the sample still had overall relatively low accuracy during RP (43%) and, if the errorful generation effect played a role here, we would have expected to see an RP benefit in these participants.”*

Reviewer 2

1. As this is a pre-registered research and I have already evaluated the research plans in the first round, so without repeating what was described there in detail, I can easily articulate my main problem with the study. Earlier I indicated that a single round of presentation on 54 triplets and then a single round of retrieval practice I did not find reassuring in terms of examining the testing effect and especially the transfer effect of retrieval practice. My main problem even

then was that although there are results in the testing effect literature that have a testing effect even with low retrieval success, it has been repeatedly shown that the effect is less robust and sometimes reversed testing effect (better memory performance in the re-exposure condition) was found. The authors nevertheless decided not to change the design. The results confirmed my preliminary fears. Recall performance during practice is quite low, if I interpret the data in Table 1 well, the recall success rate of the event triplets was 43% in the practice phase, which was accompanied by a fairly high false event recall (37%). Together, we can say that the subjects were able to recall less than half of the event information correctly and provided wrong event information in every third recall trials. Since we are talking here about an episodic memory test built on triplet learning, low retrieval success combined with a significant number of false recalls cannot be considered an optimal situation to analyse the long-term effects of retrieval practice.

This is also reflected in the final results. As a testing effect, the literature primarily refers to the long-term advantage of retrieval practice over simple re-exposure in the literature, not the long-term advantage over no-exposure. It has been demonstrated countless times that retrieval practice produces a robust long-term memory advantage over re-exposure, so its lack is presumably due to the design problems discussed earlier. Since retrieval practice does not show an advantage over re-exposure in the final test, it is questionable that any difference between the two conditions would be expected in the transfer test. This problem, I think, significantly reduces the clarity of the results. Notably, it is difficult to say that the results reveal a boundary condition related to transfer, or only draw attention to the fact that low retrieval success during retrieval practice, combined with a significant amount of false recall, eliminates the testing effect in both direct recall and in the transfer effect. I see two ways to remedy this problem. 1) running a new experiment on a modified design, taking into account the consequences of this study. 2) writing a detailed limitation paragraph in the discussion where the problems I have described are explored.

Thank you for your comments. It is important to note that participants did not complete a single round of RP. They completed two, as each association was tested in both directions. In the first block, for example, they were tested on *spider – circus* and then *circus – spider* in the second block. However, we appreciate that these were not spaced RP sessions. We have added a discussion of the impact of these methodological factors, which we agree will be valuable for future studies that seek to extend this work and better understand the conditions under which RP effects emerge (and when they do not). As this is a Stage 2 manuscript (and in line with the editor’s comments) we will not be running a follow-up experiment here.

p23: *“Although participants did practice retrieval of each word-pair in an event twice in total, spaced and further repeated RP may have maximised the chances of finding an RP effect, in line with previous work that suggests the testing effect increases with multiple RP trials (e.g. Baddeley et al., 2019; Pajkossy et al., 2019; Racsomány et al., 2020). However, and as noted in our earlier justification of the methodology, the RP effect has been found to be robust even with only a single trial (Adesope et al., 2017).”*

We have added discussion of the issue of low performance, which also tied in with another comment raised by Reviewer 1 (see their comment #3).

We completely agree that the lack of transfer effect between RP and re-exposure is difficult to interpret in the context of the lack of RP effect compared to re-exposure, and mention of this is already included throughout the Discussion.

Editor

1. Please also ensure that all the below editorial sections are included where appropriate -- if any section is not applicable to your manuscript, please can we ask you to nevertheless include the heading, but explicitly state that the heading is inapplicable

We have now included the missing headings and statements in the first two pages of the manuscript. Data accessibility is located within the manuscript already before the start of the Results section, and the Funding and Acknowledgement statements were already at the beginning of the manuscript before the Abstract.

p.1: *“Ethics statement: All participants provided informed consent prior to participating. The study was approved by the Department of Psychology’s research ethics committee at the University of York (ref: 875).*

Competing interests: We declare we have no competing interests.

Authors’ contributions: JP and AH conceived of the study, designed the study, and drafted the manuscript. JP created the materials and statistical analyses, completed data collection, and analysed the data. LH contributed to the design of the study and critically revised the manuscript. AH coordinated the study. All authors gave final approval for publication and agree to be held accountable for the work performed therein.”